# Q-Router: Agentic Video Quality Assessment with Expert Model Routing and Artifact Localization

## Abstract

Video quality assessment (VQA) is a fundamental computer vision task that aims to predict the perceptual quality of a given video in alignment with human judgments. Existing performant VQA models trained with direct score supervision suffer from **(1)** *poor generalization* across diverse content and tasks, ranging from user-generated content (UGC), short-form videos, to AI-generated content (AIGC), **(2)** *limited interpretability*, and **(3)** *lack of extensibility* to novel use cases or content types. We propose Q-Router, an agentic framework for universal VQA with a multi-tier model routing system. Q-Router integrates a diverse set of expert models and employs vision–language models (VLMs) as interactive and adaptive routers that dynamically reason then ensemble the most appropriate experts conditioned on the input video semantics. We build a multi-tiered routing system based on the computing budget, with the heaviest tier involving a specific spatiotemporal artifacts localization for interpretability. This agentic design enables Q-Router to combine the complementary strengths of specialized experts, achieving both flexibility and robustness in delivering consistent performance across heterogeneous video sources and tasks. Extensive experiments demonstrate that Q-Router matches or surpasses state-of-the-art VQA models on a variety of benchmarks, while substantially improving generalization and interpretability. Moreover, Q-Router excels on the quality-based question answering benchmark, Q-Bench-Video, highlighting its promise as a foundation for next-generation VQA systems. Finally, we show that Q-Router capably localize spatiotemporal artifacts, showing potential as a reward function for post-training video generation models.

## 1 Introduction

Video quality assessment (VQA) seeks to predict perceived video quality in agreement with human perceptual judgments. The developments of large-scale video quality datasets such as LIVE-VQC (Sinno & Bovik, 2018b;a; 2019), KonViD-1k (Hosu et al., 2017b;a), YouTube-UGC (Wang et al., 2019), LSVQ (Ying et al., 2021b;a) that provide diverse, realistic video content and reliable MOS annotations covering a wide range of content distributions, have fueled the advancements of end-to-end deep learning VQA models based on ConvNets (He et al., 2016; Xie et al., 2017; Simonyan & Zisserman, 2014) or Transformers (Yuan et al., 2021; Liu et al., 2021; Tu et al., 2022) that directly learn to predict visual quality from raw pixels. Advanced neural architectures have been explored, such as the spatial-temporal attention (Kim et al., 2018; Martinez et al., 2019; Chen et al., 2021a), multi-scale feature fusion (Xu et al., 2021; Chen et al., 2021a), and spatially-sparse attention (Wu et al., 2023b; 2022; 2023a) to capture complex spatiotemporal quality aspects.

However, a growing diversity of internet video content, from studio-made to user-generated content (UGC), to what is being predicted as the next wave of AI-generated content (AIGC), makes building a robust, generalizable VQA model increasingly challenging and costly. Firstly, they often struggle to generalize under significant distribution shifts; for example, a VQA model trained on UGC has been found to suffer a substantial performance drop (around 70% drop in PLCC for COVER He et al. (2024) from UGC to AIGC) when evaluated on AIGC datasets with different distortion patterns. Secondly, existing models trained in an end-to-end manner lack *interpretability*, although some works have explored interpretable approaches to some extent (Wu et al., 2023d; He

et al., 2024; Wu et al., 2023c). This lack of diagnostic capability not only limits operational troubleshooting but also undermines trust in automated assessments. Lastly, the tightly coupled nature of most end-to-end designs makes extensibility cumbersome: adding a new feature, incorporating a novel distortion type, or adapting to emerging content classes often requires costly retraining or architectural redesign, making maintenance unscalable.

To address these challenges, we propose **Q-Router**, the first-of-its-kind agentic VQA system with expert routing and artifact localization for generic video content. Specifically, we leverage a vision-language model (VLM) to analyze the video content, then reason over a routing pool of six state-of-the-art VQA expert models to generate a structured routing plan. Based on this plan, selected expert models are invoked to produce quality predictions, and then fused based on weights specified by the routing strategy by VLM. Together, these results are compiled into a comprehensive quality report that includes aggregated scores, per-expert breakdowns, and interpretable visual evidence to produce either a final quality score for VQA tasks or an answer for VQA visual questioning tasks, offering both quantitative evaluation and actionable diagnostic insights. Q-Router offers three key advantages: (i) adaptability, by dynamically selecting relevant experts according to content and distortion characteristics; (ii) interpretability, by producing explicit routing rationales and diagnostic outputs; and (iii) robustness, by ensuring reliable performance across heterogeneous video domains, including both user-generated and AI-generated content.

To account for various inference budgets, we introduce three different tiers for Q-Router: ❶ Tier 0: a lightweight baseline that invokes a single expert model from the routing pool, providing fast yet coarse quality assessment; ❷ Tier 1: the standard configuration that the VLM performs coarse-grained reasoning and then assigns adaptive weights to experts accordingly; ❸ Tier 2: the full spatiotemporal artifact localization pipeline, which fuses expert predictions into a robust quality score while simultaneously identifying frame regions responsible for distortions, providing interpretable diagnostic evidence. The multi-tiered design enables Q-Router to support tasks from lightweight benchmarking to in-depth diagnostic analysis. Extensive experimental results demonstrate that Q-Router consistently achieves state-of-the-art performance across both standard VQA benchmarks and quality-related visual question answering tasks, validating its effectiveness, robustness, and versatility in diverse evaluation settings.

Our contributions are summarized as follows:

- We present Q-Router, a first-of-a-kind agentic VQA framework that leverages a vision-language model to conduct structured reasoning and dynamic expert routing.
- We design a multi-tier system for Q-Router that adapts to different inference budgets, supporting tasks from lightweight benchmarking to in-depth diagnostic analysis.
- We introduce a spatial-temporal artifact localization pipeline that pinpoints distortions, offering novel interpretability and critical diagnostic evidence.
- We conduct comprehensive experiments on both UGC and AIGC video domains, showing that Q-Router consistently outperforms previous state-of-the-arts in terms of accuracy, robustness, and interpretability, across both video quality assessment and visual question answering tasks.

## 2 METHODOLOGY

In this section, we present Q-Router, an agentic routing framework designed for diverse video quality assessment (VQA) tasks. The framework provides a unified paradigm that integrates diverse expert models with cost-aware routing and fusion mechanisms. By decoupling routing policies, expert specialization, and integration strategies, Q-Router enables the flexible composition of different techniques, yielding a "template" for designing general VQA systems that are efficient, scalable, transparent, and adaptable across heterogeneous video sources and assessment tasks.

### 2.1 OVERVIEW

The general procedure of Q-Router is outlined in Algorithm 1, which consists of three fundamental and complementary components.

The first component is the router $\mathcal{L}$ with strategy $\mathcal{S}$, which jointly determine how the system selects a subset of expert models from the available routing pool $\mathcal{P}$. In practice, the router can be instantiated

---

**Algorithm 1** Q-Router in General

---

**Required:**
(1) A router $\mathcal{L}$ and routing strategy $\mathcal{S}$;
(2) a routing pool $\mathcal{P}$;
(3) a fusion operator $\mathcal{F}$.

**Input:** The input video $\mathbf{v}$.

1: $(\mathcal{M}, \boldsymbol{\alpha}) \leftarrow \mathcal{S}(\mathcal{L}, \mathcal{P}, \mathbf{v})$        $\triangleright$ Select experts $\mathcal{M} = \{m_i\}_{i=1}^{k}$ and optional weights $\alpha_i$
2: **for** $m_i \in \mathcal{M}$ **do**
3:     $r_i \leftarrow m_i(\mathbf{v})$        $\triangleright$ Run VQA expert to obtain prediction/descriptor
4: $r^\star \leftarrow \mathcal{F}(\mathbf{v}, \{(r_i, \alpha_i)\}_{i=1}^{k})$        $\triangleright$ Cost-aware fusion of expert outputs
5: **return** $r_*$

---

as a VLM model, and the strategy can range from rule-based heuristics to learned policies that trade off between accuracy and computational budget. This abstraction allows the routing decision to be flexibly adapted to the task or resource constraints at hand.

The second component is the routing pool of expert models $\mathcal{P} = \{m_i\}$, each specializing in a different perspective/domain of VQA. Examples include experts focused on UGC or AIGC. Given an input video $\mathbf{v}$, the router selects an appropriate subset of experts $\mathcal{M} \subseteq \mathcal{P}$ to execute. This modular design naturally supports extensibility, since new VQA experts can be integrated into the pool without altering the routing pipeline.

The third component is the fusion function $\mathcal{F}$, which aggregates the outputs of the selected experts into a single final score $r^\star$. Importantly, $\mathcal{F}$ is cost-aware: it can incorporate not only expert predictions but also their computational cost, confidence, or reliability. For example, $\mathcal{F}$ may weight more lightweight experts higher under tight budget constraints, while leveraging stronger but expensive experts when sufficient resources are available. Furthermore, when computational resources are not a limiting factor, additional priors or auxiliary signals can be incorporated into the fusion stage to enhance performance and provide deeper insights into video quality assessment.

Altogether, Q-Router provides a flexible framework that decouples routing, expert modeling, and fusion. This separation of concerns enables efficient, transparent, and scalable video quality evaluation across diverse video sources and assessment tasks.

## 2.2 Q-ROUTER PIPELINE

As shown in Figure 1, the Q-Router system employs a three-tiered routing hierarchy that organizes routing strategy and fusion operator according to varying cost–performance trade-off requirements. In practice, this design enables the user to dynamically adjust the inference pipeline based on available computational resources and the complexity of the input video. **Tier 0** prioritizes efficiency and rapid assessment, while higher tiers (**Tier 1 & 2**) leverage more specialized experts and richer fusion mechanisms to achieve greater accuracy and interpretability. This hierarchical structure ensures that Q-Router can flexibly deploy to diverse scenarios, ranging from resource-constrained real-time applications to high-fidelity offline evaluations.

**Tier 0**    At the base level of Q-Router, the router operates in its most lightweight configuration by selecting a *single* expert model from the routing pool by considering the characteristics of the input video, such as its structural complexity, content modality, and observable quality attributes. Once the most appropriate expert is identified, the fusion operator directly adopts the prediction of this model as the Q-Router's final output. By design, this tier prioritizes computational efficiency and fast inference, making it particularly suitable for scenarios where resource constraints or real-time processing requirements outweigh the need for more complex multi-expert aggregation.

**Tier 1**    At the intermediate tier, Q-Router expands the inference pipeline by enabling the router to select multiple expert models rather than relying on a single candidate. The predictions from these experts are subsequently integrated through a relevance-aware weighted fusion operator, which dynamically adjusts weights according to the input's characteristics. This design enhances robustness

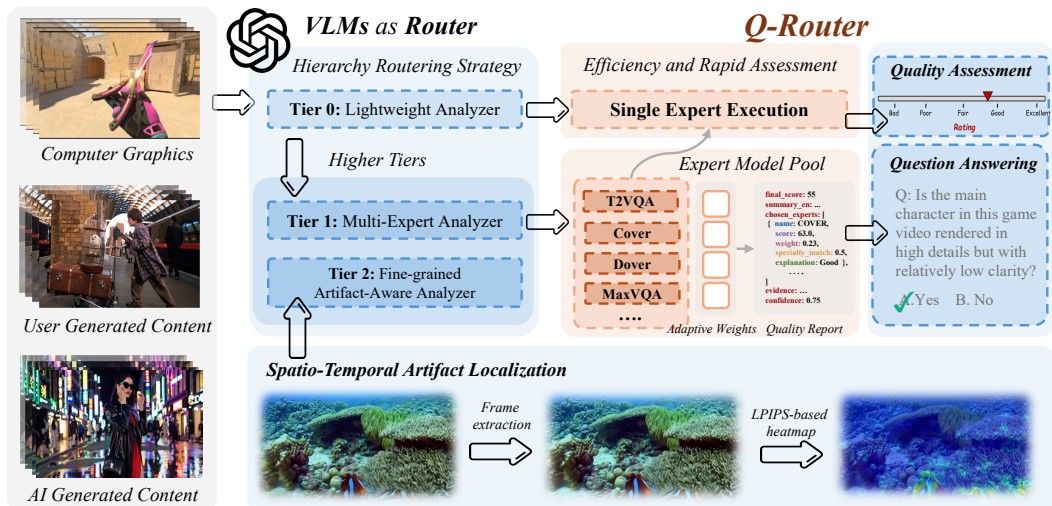

Figure 1: We present Q-Router, an agentic routing framework designed for diverse video quality assessment tasks. Q-Router leverages a VLM as the router to dynamically assign the most suitable expert model from a comprehensive pool of state-of-the-art VQA methods. The expert pool includes COVER (He et al., 2024), DOVER (Wu et al., 2023d), BVQA (Wen et al., 2024), UVQA (Wang et al., 2021), MaxVQA (Wu et al., 2023c), and T2VQA (Kou et al., 2024), enabling robust and adaptive evaluation across user-generated, AI-generated, and computer-generated video content.

by leveraging complementary judgments across diverse experts, thereby improving both stability and predictive accuracy compared to **Tier 0**. Furthermore, the adaptive weighting mechanism provides a principled approach to performance optimization, allowing the system to balance accuracy with efficiency under varying content and quality conditions.

**Tier 2** At the advanced tier, Q-Router augments the inference pipeline by explicitly incorporating artifact localization into the fusion process. In this configuration, the router selects multiple expert models, while the fusion operator integrates spatially localized artifact maps that identify regions of potential quality degradation within frames or clips exhibiting artifacts. By leveraging these fine-grained signals, the system guides the fusion process toward distortion-prone areas, thereby enhancing both the accuracy and interpretability of the final assessment. This design not only strengthens robustness in challenging scenarios—such as AI-generated or heavily compressed content—but also grounds predictions in spatial evidence of distortions. By combining expert diversity with region-aware fusion, **Tier 2** delivers the highest level of reliability and diagnostic value within the Q-Router hierarchy. Moreover, the produced artifact maps can provide actionable insights supporting the post-processing tasks, such as targeted video restoration.

## 2.3 SPATIO-TEMPORAL ARTIFACT LOCALIZATION

In this section, we detail the process of conducting spatio-temporal artifact localization for Tier 2 of Q-Router (more details can be found in Appendix A.3). Unlike Tier 0 and Tier 1, which operate solely on expert routing and aggregated scores, Tier 2 introduces fine-grained artifact localization across both spatial and temporal dimensions. This addition not only enhances the accuracy of video quality assessment but also provides interpretable evidence of model predictions.

**Probabilistic Frame Extraction** Videos are first processed by the probabilistic extractor, which adaptively samples frames most likely to contain perceptual artifacts. Using handcrafted features (motion residuals, sharpness energy, gradient kurtosis, edge density, color histogram shifts, and optional semantic priors), each frame is assigned an artifact probability through a weighted logistic model. High-probability frames are grouped into temporally coherent clips via hysteresis thresholding and further refined by diversity sampling to ensure comprehensive coverage while remaining computationally efficient.

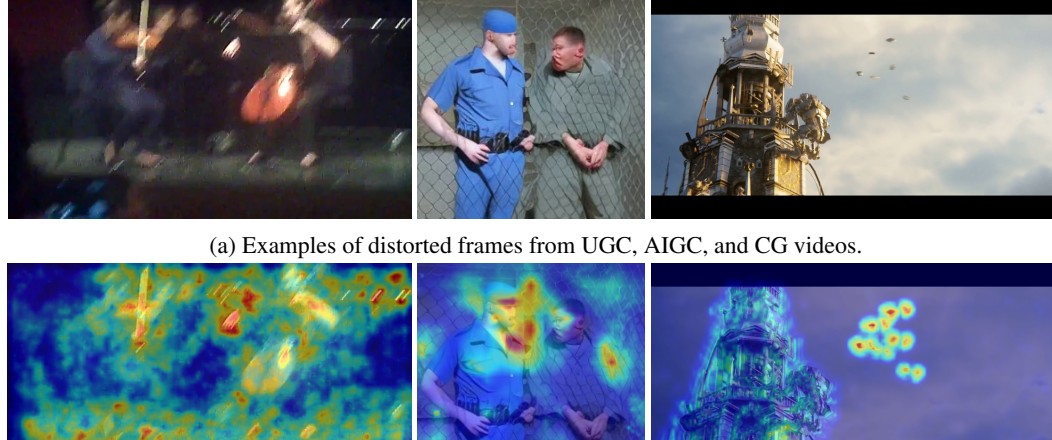

(a) Examples of distorted frames from UGC, AIGC, and CG videos.

(b) Corresponding artifact localization heatmaps for the distorted frames.

Figure 2: Distorted frames and their corresponding artifact localization heatmaps across UGC, AIGC, and CG videos. The first row shows example distorted frames, while the second row highlights suspicious regions detected by Q-Router.

**VLM-Based Artifact Filtering** To further improve the reliability of artifact detection, candidate frames from above frame extraction step are filtered using a GPT-4o model. Each frame is categorized into one of three artifact classes: (1) visual hallucinations (e.g., implausible objects or persons), (2) image artifacts (e.g., compression distortions, blurring, pixelation), or (3) AI-generated inconsistencies (e.g., unrealistic lighting or anatomy). Only frames flagged as containing artifacts are retained for localization, while clean frames are excluded.

**Perceptual Heatmap Generation and Severity Estimation** Within each retained clip, consecutive frame pairs are compared using the LPIPS (Zhang et al., 2018) metric. To compensate for temporal motion, the subsequent frame is first aligned to the reference frame via optical flow (Farnebäck). The resulting LPIPS activation maps capture fine-grained perceptual differences and are normalized into interpretable heatmaps, as illustrated in Figure 2. Clip-level severity is then quantified by aggregating heatmap intensities via the mean pooling strategy, and the frame pair with the highest severity score is selected as the representative artifact instance for that clip. Finally, the corresponding heat map of the representative pair for each clip would be provided as an augmentation for Q-Router during the evaluation of video quality.

## 3 EXPERIMENTS

We conduct two categories of experiments, corresponding to distinct VQA scenarios—quality rating and question answering—to validate the effectiveness of the proposed method. First, we evaluate Q-Router on classic VQA scoring datasets containing both UGC and AIGC videos. The results show that our method achieves consistently robust performance across benchmarks, outperforming baselines and achieving the best average results overall. Next, we assess the effectiveness of Q-Router on video quality question-answering tasks, with a focus on the Q-Bench-Video benchmark. On this benchmark, our approach establishes a new state of the art, *ranking at the top of the public leaderboard*.

### 3.1 VIDEO QUALITY ASSESSMENT

In this section, we evaluate Q-Router on video quality assessment tasks where the objective is to rate the perceptual quality of input videos. This setting reflects the classical formulation of VQA, in which baseline models are pre-trained to predict quality scores that align with human subjective judgments.

**Datasets**   We conduct experiments on both UGC and AIGC VQA tasks. Specifically, for UGC, we evaluate our proposed method on widely used benchmarks including KoNViD-1k (Hosu et al., 2017b;a), LSVQ-1080 (Ying et al., 2021b;a), LSVQ-Test (Ying et al., 2021b;a), LIVE-VQC (Sinno & Bovik, 2018b;a; 2019), and Youtube-UGC (YT-UGC) (Wang et al., 2019). For AIGC, we assess performance on the T2VQA-DB (Kou et al., 2024) dataset, which focuses on text-to-video generated content and poses unique challenges distinct from conventional UGC benchmarks.

**Baselines**   We benchmark Q-Router against several state-of-the-art models for UGC VQA, including COVER (He et al., 2024), DOVER (Wu et al., 2023d), MaxVQA Wu et al. (2023c), ModularB-VQA (Wen et al., 2024), and UVQ (Wang et al., 2021), which represent diverse design paradigms for no-reference VQA. For AIGC VQA, we adopt T2VQA (Kou et al., 2024) as the baseline, which is developed on the T2VQA-DB dataset specifically for evaluating text-to-video generated content.

| Dataset | Metric | COVER | DOVER | MaxVQA | BVQA | T2VQA | UVQ | Q-Router |
|---------|--------|-------|-------|--------|------|-------|-----|----------|
| *UGC Video Quality Assessment Benchmark* | | | | | | | | |
| KoNViD-1k | PLCC | 0.8528 | 0.8447 | 0.8627 | **0.8731** | 0.1717 | 0.6998 | 0.8612 |
|           | SRCC | 0.8507 | 0.8341 | 0.8599 | **0.8713** | 0.1510 | 0.7067 | 0.8607 |
| LSVQ-1080 | PLCC | 0.7806 | 0.7811 | 0.7419 | 0.7337 | 0.2371 | 0.5304 | **0.8076** |
|           | SRCC | 0.7544 | **0.7808** | 0.7282 | 0.7431 | 0.2479 | 0.5068 | 0.7792 |
| LSVQ-Test | PLCC | 0.8378 | **0.8660** | 0.8005 | 0.8583 | 0.2368 | 0.6801 | 0.8540 |
|           | SRCC | 0.8465 | **0.8777** | 0.8100 | 0.8514 | 0.2460 | 0.6858 | 0.8585 |
| LIVE-VQC | PLCC | 0.7624 | 0.8742 | 0.7325 | 0.8571 | 0.1719 | 0.6827 | **0.8806** |
|          | SRCC | 0.7254 | 0.8187 | 0.6942 | 0.8103 | 0.2015 | 0.6800 | **0.8293** |
| YT-UGC | PLCC | 0.1529 | 0.7367 | 0.7761 | 0.6974 | 0.2901 | **0.9402** | 0.8920 |
|        | SRCC | 0.1413 | 0.7195 | 0.7789 | 0.6749 | 0.2777 | **0.9457** | 0.8909 |
| *AIGC Video Quality Assessment Benchmark* | | | | | | | | |
| T2VQA-DB | PLCC | 0.2468 | 0.0642 | 0.2361 | 0.1051 | 0.7733 | 0.1158 | **0.8283** |
|          | SRCC | 0.1377 | 0.0429 | 0.2010 | 0.0691 | 0.7828 | 0.0983 | **0.8258** |
| *Overall Performance* | | | | | | | | |
| Average | PLCC | 0.6056 | 0.6945 | 0.6916 | 0.6874 | 0.3135 | 0.6081 | **0.8539** |
|         | SRCC | 0.5760 | 0.6789 | 0.6787 | 0.6700 | 0.3178 | 0.6038 | **0.8407** |

Table 1: Comparison of methods on UGC and AIGC benchmarks against the Q-Router (Tier 1). Best results are highlighted in bold, while the second-best results are underlined.

**Experimental Setup**   We employ GPT-4o (Hurst et al., 2024; OpenAI, 2024) as the backbone router and fusion operator within Q-Router. The router is responsible for analyzing the input video descriptors and selecting appropriate expert models, while the fusion operator integrates their outputs into a unified quality report and produces the final quality score on a $[0, 100]$ scale. For reproducibility, the complete prompt used to deploy GPT-4o, including routing instructions and fusion templates, is provided in Appendix A.1.

As for the routing pool, we include all the baseline models evaluated in this section, which collectively serve as the candidate experts. For UGC VQA, we leverage the publicly released model weights to ensure faithful implementation and evaluation. Since T2VQA does not provide open-sourced model weights, we reproduce its training process by following the official repository[1].

**Results**   Table 1 summarizes the performance of Q-Router compared with state-of-the-art baselines on both UGC and AIGC VQA benchmarks. Our method achieves the best average performance across datasets, with a PLCC of $0.8539$ and SRCC of $.8407$, outperforming all individual baselines by a significant margin over $20\%$, highlighting the effectiveness of the routing-and-fusion paradigm in leveraging complementary strengths of heterogeneous expert models.

---

[1]https://github.com/QMME/T2VQA

On UGC datasets, Q-Router consistently demonstrates superior or highly competitive results. For example, it achieves the best PLCC (0.8076) on LSVQ-1080 and the second-best SRCC (0.8607) on KoNViD-1k, surpassing strong baselines such as COVER (He et al., 2024), DOVER (Wu et al., 2023d), and MaxVQA (Wu et al., 2023c). Importantly, Q-Router maintains robustness across different datasets, whereas individual baselines often exhibit dataset-specific strengths but lack generalization.

On the T2VQA-DB dataset, Q-Router significantly outperforms all baselines, reaching a PLCC of 0.8283 and SRCC of 0.8258. This demonstrates the Q-Router's ability to adaptively leverage expert knowledge and handle distribution shifts in AI-generated video quality—a setting where conventional UGC-trained models fail (PLCC $< 0.25$).

## 3.2 VISUAL QUESTION ANSWERING ON VIDEO QUALITY

**Datasets**  We further conduct experiments on visual question answering tasks related to video quality. Specifically, we evaluate a wide range of VLMs, along with Q-Router, on Q-Bench-Video (Zhang et al., 2025b), which is the first and currently only benchmark dedicated to this task.

**Baselines**  We deploy Q-Router with GPT-4o (Hurst et al., 2024; OpenAI, 2024) as the backbone, and evaluate it alongside several widely used open-sourced VLMs, including LLaVA-Next (Li et al., 2024b), LLaVA-v1.5 (Liu et al., 2024a), mPLUG-Owl2 (Ye et al., 2024b), mPLUG-Owl3 (Ye et al., 2024a), LLaVA-OneVision (Zhang et al., 2025a), InternVL-Chat (Chen et al., 2024), VILA1.5 (Ke et al., 2023), PLLaVA (Xu et al., 2024), LLaVA-Next-Video (Li et al., 2024b), ST-LLM (Liu et al., 2024b), Video-LLaVA (Lin et al., 2023), VideoChat2 (Li et al., 2023b), GPT-4o (Hurst et al., 2024), and VQA[2] (Jia et al., 2025). These models represent the current generation of multimodal models designed for video understanding and serve as competitive baselines. In addition, we include GPT-4o itself as a strong proprietary baseline, as it currently ranks as the state-of-the-art model on the Q-Bench-Video leaderboard (Zhang et al., 2025b).

**Experimental Setup**  We adopt the same experimental settings as described in Section 3.1 for video quality rating, ensuring consistency across evaluation protocols. Specifically, we employ GPT-4o (Hurst et al., 2024) as the backbone router and fusion operator within Q-Router, and construct the routing pool using the same set of expert models introduced earlier. The evaluation is conducted on Q-Bench-Video (Zhang et al., 2025b), a large-scale benchmark designed for video quality question answering, which covers diverse question types (e.g., Yes-or-No, What/How), video modalities (UGC, AIGC, and CG), and contextual reasoning settings (e.g., Referring, Global, and Comparison). To ensure reproducibility, we follow the official evaluation protocol of Q-Bench-Video and report accuracy across all categories as well as the overall score.

**Results**  Table 2 presents the performance of Q-Router on the Q-Bench-Video `dev` split, compared against a set of strong VLM baselines. Our proposed routing framework achieves consistent improvements across all tiers. Tier 0, which routes to a single expert, already matches GPT-4o (56.59 vs. 56.91 overall), while substantially outperforming it on AIGC-related questions (56.67 vs. 44.25). Tier 1, which leverages weighted expert fusion, further improves to 59.40 overall, with strong gains in technical (59.48) and aesthetic (62.90) dimensions. Finally, Tier 2, which integrates artifact localization into the fusion stage, delivers the best results overall (60.07), surpassing GPT-4o. Tier 2 achieves new state-of-the-art performance across most categories, notably Yes-or-No (76.00), What/How (57.01), open-ended (43.33), and temporal concerns (61.46).

## 3.3 ABLATION STUDY

In this section, we conduct the ablation study on the key component of our proposed Q-Router framework. Specifically, we examine the performance of Q-Router under different prompting structures:, including **(1)** *Zero-shot Prompting*: the VLM is directly prompted to output the predicted rating for the input video on a scale of $[0, 100]$; **(2)** *CoT prompting*: The prompt template includes background information on the dimensions and aspects that should be considered when assessing the video quality; **(3)** *Expert Model Fusing*: directly prompt the VLM to fuse the results from expert models.; **(4)** the comprehensive *Q-Router Tier 1* framework. All experiments in this section are

| Method | Overall | Question type | | | Quality Concern | | | |
|---|---|---|---|---|---|---|---|---|
| | | Yes-or-No | What/How | Open-ended | Technical | Aesthetic | Temporal | AIGC |
| LLaVA-Next | 47.00 | 63.20 | 43.78 | 30.42 | 45.95 | 54.83 | 45.63 | 46.24 |
| LLaVA-v1.5 | 45.57 | 53.40 | 46.87 | 33.85 | 55.83 | 55.90 | 44.91 | 45.96 |
| mPLUG-Owl2 | 44.20 | 59.61 | 38.83 | 31.57 | 42.49 | 53.28 | 44.73 | 40.07 |
| mPLUG-Owl3 | 52.44 | 60.82 | 56.52 | 35.84 | 51.34 | 60.46 | 54.26 | 37.30 |
| LLaVA-OneVision | 52.12 | 62.13 | 52.23 | 38.56 | 48.74 | 61.53 | 48.81 | 44.57 |
| InternVL-Chat | 51.91 | 70.21 | 48.65 | 32.20 | 50.24 | 49.50 | 52.96 | 47.69 |
| VILA1.5 | 49.62 | 61.59 | 47.30 | 36.88 | 46.74 | 59.30 | 47.57 | 43.67 |
| PLLaVA | 51.23 | 65.13 | 54.23 | 29.44 | 50.31 | 60.09 | 50.13 | 50.75 |
| LLaVA-Next-Video | 48.97 | 65.98 | 45.31 | 31.92 | 48.11 | 57.33 | 47.09 | 45.56 |
| ST-LLM | 35.89 | 46.43 | 28.45 | 32.31 | 33.32 | 45.66 | 36.01 | 32.62 |
| Video-LLaVA | 45.89 | 64.36 | 39.38 | 30.86 | 43.35 | 55.97 | 45.58 | 43.64 |
| VideoChat2 | 42.06 | 56.54 | 33.13 | 35.36 | 39.09 | 49.27 | 41.59 | 38.04 |
| GPT-4o[†] | 56.91 | 69.95 | 55.20 | 42.10 | 55.74 | **66.06** | 57.66 | 44.25 |
| VQA[2] | 56.50 | 75.42 | 56.93 | 37.78 | **61.04** | - | 60.36 | - |
| Q-Router *(Tier 0)* | 56.59 | 74.82 | 54.30 | 35.80 | 54.86 | 51.34 | 54.91 | **56.67** |
| Q-Router *(Tier 1)* | 59.40 | 75.53 | 56.33 | 42.44 | 59.48 | 62.90 | 58.45 | 45.00 |
| Q-Router *(Tier 2)* | **60.07** | **76.00** | **57.01** | **43.33** | 59.31 | 65.63 | **61.46** | 50.22 |

Table 2: Comparison of methods on the closed-ended questions in Q-Bench-Video `dev` split across question types and quality concerns. The best results are highlighted in bold, while the second-best results are underlined. [†]: As the models continue to evolve, we reproduce and report the results of the GPT-4o model for comparison.

conducted using the Q-Router (Tier 1) configuration and GPT-4o as the backbone. All experiments use the Tier 1 configuration with GPT-4o as the backbone, following the same setup as Section 3.2.

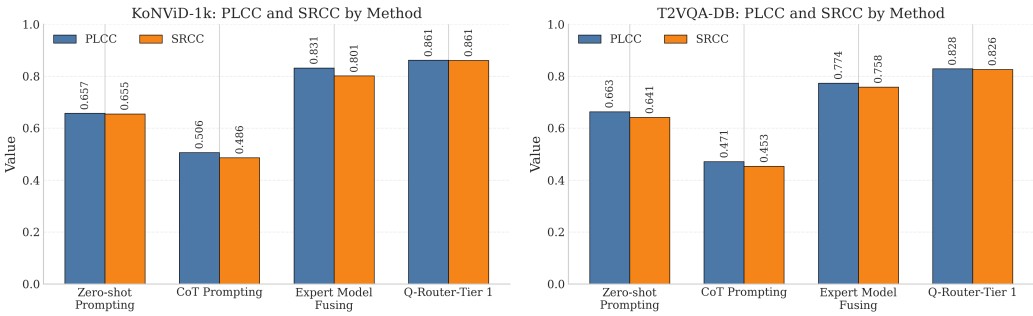

Figure 3: Impact of prompting structure across both UGC and AIGC benchmarks for VQA, with GPT-4o as backbone.

As presented in Figure 3, zero-shot prompting results exhibit surprisingly high correlation with MOS, reflecting the effectiveness of VLM's system 1 thinking on VQA tasks. While CoT prompting performs substantially worse, suggesting that introducing forced deliberation through chain-of-thought may disrupt alignment with human perception in the VQA setting. Additionally, the expert model fusing strategy achieves a notable performance boost, confirming the complementary strengths of multiple specialized VQA models. Finally, our proposed Q-Router (Tier 1) yields the highest performance across both benchmarks, indicating that the adaptive routing framework not only leverages the expert knowledge but also preserves alignment with human preference.

## 4 RELATED WORKS

**Video Quality Assessment** Traditional approaches to video quality assessment (VQA) largely rely on full-reference (FR) and reduced-reference (RR) metrics, such as PSNR, SSIM (Wang et al., 2004), and VMAF (Li et al., 2016). To address the reliance on reference videos for FR, no-reference (NR) models were introduced, estimating quality directly from distorted videos by extracting hand-

crafted features related to motion, texture, and temporal consistency (Saad et al., 2014; Netflix, 2016). However, the generalization ability of handcrafted features remains limited, particularly for diverse real-world content. The advent of deep learning has substantially advanced NR-VQA. Early CNN-based models focused on learning spatial representations (Kim et al., 2018), while later works incorporated temporal modeling (Chen et al., 2021b; Li et al., 2019). Large-scale benchmarks such as KoNViD-1k (Hosu et al., 2017b), LSVQ (Ying et al., 2021b), and YouTube-UGC (Wang et al., 2019) have provided the foundation for training and evaluating deep VQA models, fostering rapid progress in the field. Recent works have explored modular and expert-driven designs for VQA. Methods such as COVER (He et al., 2024), DOVER (Wu et al., 2023d), and MaxVQA (Wu et al., 2023c) leverage multi-branch architectures or ensembles to account for diverse perceptual factors. However, most existing approaches adopt fixed fusion strategies, limiting adaptivity across heterogeneous inputs. In contrast, our work introduces Q-Router, a routing framework that leverages VLMs to dynamically select and fuse expert predictions, offering a flexible and extensible solution that adapts to diverse video inputs while maintaining interpretability.

**LLM Reasoning and Agent**  Large Language Models (LLMs) (Devlin et al., 2018; Radford et al., 2019; Brown et al., 2020; Team et al., 2023; Roziere et al., 2023; Touvron et al., 2023a;b; Raffel et al., 2020; Yang et al., 2024; Team, 2024; Pan et al., 2024) have demonstrated remarkable capabilities on complex reasoning and understanding tasks across natural language, code generation, and multimodal domains. Reasoning techniques/framework like Chain-of-Thought (CoT) prompting (Wei et al., 2022) and self-consistency (Wang et al., 2022) have been shown to significantly enhance reasoning performance by encouraging models to generate intermediate steps rather than producing direct answers. More recent works explore decompositional reasoning (Zhou et al., 2022) and tool-augmented reasoning (Schick et al., 2023), highlighting the ability of LLMs to integrate the structured process and external knowledge.

Building on these advances, Large Vision Language Models (VLMs) (Li et al., 2022; 2023a; Liu et al., 2024a; Li et al., 2024b; Meta, 2024; Bai et al., 2023; Wang et al., 2024; Lu et al., 2024; Wu et al., 2024; Bai et al., 2025) extended the understanding and reasoning capabilities of LLMs into the visual domain. Recent works demonstrate that VLMs can perform not only captioning or classification, but also higher-level reasoning over visual inputs in real-world application scenarios (Moor et al., 2023; Li et al., 2024a; Shao et al., 2024; Tian et al., 2024; Sima et al., 2023; Xing et al., 2025a; Ma et al., 2025; Wang et al., 2025; Rana et al., 2023; Kim et al., 2024; Xing et al., 2025b).

A growing line of work has begun to frame LLMs and VLMs as agents capable of orchestrating modular pipelines. Prior studies demonstrate that such models can act as routers, dynamically selecting and invoking specialized tools or experts to solve complex tasks Schick et al., 2023; Yao et al., 2023; Sumers et al., 2023; Shen et al., 2023; Luo et al., 2025; Zuo et al., 2025. These agentic approaches emphasize the potential of LLMs/VLMs not merely as standalone predictors but as controllers of adaptive, extensible systems—a paradigm shift that directly inspires our design of Q-Router as a routing-based framework for video quality assessment.

## 5 DISCUSSIONS

**Performance of Expert Fusion**  As the experiments demonstrate in Table 1, the Q-Router can not consistently achieve the best performance compared to the expert models in its pool on each single dataset. However, Q-Router's goal is to achieve robust and interpretable performance across video domains and task types, rather than to guarantee that fusion always exceeds the strongest expert on every individual benchmark.

While, in the ideal case, we expect Q-Router to outperform the best expert in its pool, this is not a theoretically guaranteed condition for VLM-based systems—especially when the expert pool contains highly specialized models optimized for particular datasets. Therefore, our primary goal is to provide a generalizable and interpretable test-time framework that performs consistently well across video domains and task types. As shown in Table 1, Q-Router achieves competitive performance on each single benchmark and achieves the best average performance. This trend becomes even more pronounced in Table 2 on the Q-Bench-Video benchmark, where Q-Router achieves the state-of-

| Method | COVER | DOVER | MaxVQA | BVQA | UVQ | T2VQA | Q-Router-T0 | Q-Router-T1 | Q-Router-T2 |
|---|---|---|---|---|---|---|---|---|---|
| **Time (s)** | 2.09 | 0.10 | 0.17 | 5.35 | 13.16 | 0.28 | 7.11 | 8.46 | 16.36 |

Table 3: Inference latency measured in seconds per video. Q-Router timing reflects only routing overhead (VLM inference + preprocessing).

the-art performance, highlighting the effectiveness of our multi-expert routing and fusion strategy in settings with diverse or mixed tasks.

**Design of the Artifacts Localization Module**   The primary goal of Q-Router is to serve as a test-time scaling framework for VQA tasks, rather than a fully end-to-end training-based method. And it prioritizes robustness, interpretability, and generalizability across diverse VQA task settings.

Our handcrafted feature–driven artifact localization method can offer stable and interpretable results that are independent of the underlying VQA models. Additionally, to the best of our knowledge, the end-to-end learned diagnostic methods that can effectively perform fine-grained, distortion-level artifact localization are still in the early stage of research. This motivates our choice of a feature-based Tier-2 design, which can ensure robust performance across diverse real-world scenarios while remaining compatible with the broader test-time scaling framework of Q-Router.

**Computational Overhead**   We provide a detailed analysis of the inference latency in Table 3, reported as seconds per video. To clearly quantify the computational overhead introduced by our framework, the time costs for Q-Router (Tier 0, Tier 1, and Tier 2) reflect only the routing-related components—including VLM inference and frame preprocessing—and *exclude* the execution time of the expert VQA models (e.g., COVER, DOVER), which are listed separately for comparison. As shown in the table, lightweight expert models such as DOVER (0.10s) and MaxVQA (0.17s) achieve very low latency, whereas Q-Router introduces additional computational cost due to its multi-stage routing and diagnostic analysis. This overhead, however, is necessary to provide the fine-grained interpretability, cross-domain robustness, and diagnostic reliability that scalar VQA models alone cannot offer.

## 6 CONCLUSION

In this work, we introduced Q-Router, an agentic framework for video quality assessment that leverages VLMs as routers and fusion operators over a diverse pool of expert models. By structuring the inference process into a three-tier hierarchy—ranging from lightweight single-expert routing, to multi-expert weighted fusion, and finally to artifact-localized fusion—Q-Router achieves a balance between efficiency, robustness, and interpretability.

Extensive experiments on both classical VQA benchmarks (UGC and AIGC) and the Q-Bench-Video question-answering benchmark demonstrate that Q-Router consistently outperforms state-of-the-art baselines. Beyond improving performance, Q-Router enhances interpretability by grounding outputs in spatial evidence of distortions and offers actionable insights that can inform downstream tasks like restoration and adaptive post-processing. These results highlight the promise of expert routing as a paradigm for advancing video quality assessment. Furthermore, the modular nature of Q-Router provides opportunities for extending the framework to broader multimodal evaluation tasks, incorporating new expert models as they emerge, and exploring more advanced routing strategies that further improve efficiency and generalization.

# 7 ETHICS STATEMENT

Our work on Q-Router adheres to the ICLR Code of Ethics[2] . The research does not involve human subjects, personally identifiable information, or sensitive attributes, and therefore does not raise concerns related to human experimentation, privacy, or consent. All datasets used are publicly available under appropriate licenses, and no proprietary or confidential data were employed.

# 8 REPRODUCIBILITY STATEMENT

We comprehensively presnet the implementation details in Sectio 2.2, Appendix A.3, and A.1. All the evaluation code are available from public online code repositories. All data and source code of Q-Router pipeline will be released upon acceptance if they are not from online codebase.

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

# A  APPENDIX

## A.1  PROMPTS USED IN THE ROUTING PROCESS

```
You are a VQA-Orchestrator, an expert agent specializing in video quality assessment and model routing.

**Your task**:

1. Given the video's sampled frames (16 for single video, 8+8 for comparison) and metadata (type,
   description, orientation, etc.), first classify the video into one of:  user-generated content(UGC),
   AI-generated content (AIGC), computer graphics (CG) Videos.

2. Using the expert model cards below, select the most appropriate expert(s) based on their known
   biases, the video context, inter-model agreement, and confidence priors.  If metadata/content
   indicates AI-generated, strongly prioritize **T2VQA**.

3. Produce a strictly formatted JSON output with:

     • 'retuned_experts' (list of selected model names)
     • 'retionale' (justification referencing keyframes (e.g., blur, banding, freeze, misalignment)

**Expert model cards (for routing logic)**:

• **COVER**:  Uses three parallel branches|technical (Swin Transformer), aesthetic (ConvNet), semantic
  (CLIP encoder)|combined via a cross-gating block to capture compression artifacts, aesthetic
  composition, and semantic coherence (real-time,  96 fps) CVF Open Access.

• **DOVER++**:  Extension of DOVER family; overlaps with COVER's aesthetic/technical branches; use
  primarily as a consistency reference unless its output aligns much better with most other experts.

• **UVQ**:  Google's YouTube-trained universal VQA model built from millions of UGC examples; robust
  baseline when domain unclear or disagreements are large ResearchGate] Google Research.

• MaxVQA (ExplainableVQA)**:  CLIP-based, language-prompted model that provides both overall quality
  and fine-grained human-readable factors (e.g.  banding, blur, aesthetic issues); used only for
  explanation and weight hints, not scoring arXiv.

• **ModularBVQA**:  Lightweight, modular baseline model suitable for low-latency deployment and serving
  as a fallback; modest sensitivity to capture-induced distortions.

• **T2VQA**:  Text-to-video alignment model designed for AI-generated content, assessing fidelity
  between textual prompt and video; upweighted when video is AI-generated and textual condition is
  provided.
```

Figure 4: Prompt for selecting expert models with Q-Router using GPT-4o.

The prompt used for Q-Router Tier 1 on standard VQA tasks with GPT-4o as the backbone is shown in Figure 5. For quality-related question answering, we employ the prompts illustrated in Figures 6, 7, and 8 for the Tier 0, Tier 1, and Tier 2, respectively.

## A.2  DETAILED ON IMPLEMENTATION OF Q-ROUTER

Overall, our proposed Q-Router framework consists of three tiers, each progressively enriching the routing process:

- The Tier 0 serves as the lightweight first-stage expert VQA model selector to identify which one expert model is likely to be relevant.
- The Tier 1 employs the VLM router as a fusion operator to results from multiple expert VQA models.
- The Tier 2 further provides a fine-grained analysis of detailed artifact localization.

The complete Tier-2 routing workflow is detailed in Algorithm 2.

## A.3  DETAILS ON ARTIFACT LOCALIZATION

The Probabilistic Extractor (Algorithm 3) identifies frames most likely to contain artifacts. Seven handcrafted features are extracted, including motion residuals (temporal discontinuities), Laplacian variance (sharpness), gradient kurtosis (posterization), edge density (ringing), histogram distance (shot changes), and optional semantic priors (face, text). Features are normalized via median–IQR scaling to suppress outliers. Each frame is assigned a probability $p_t$ through a weighted logistic model, with higher weights on motion residuals, histogram distance, and gradient kurtosis. Frames above a high threshold initiate clips, which continue until probabilities drop below a low threshold.

```
You are VQA-Orchestrator, an expert-level agent for fusing multiple Video Quality Assessment (VQA)
experts into a final quality score.

**Your task**:
1. Given the video's metadata (type, description, orientation, etc.)  and expert scores (all already
   scaled to [0-100]), dynamically assign weights to each expert based on their known biases, the video
   context, inter-model agreement, and confidence priors.

2. Use these weighted scores to compute a **final_score**:

     • If 'max(score) - min(score) > 20', use weighted median; otherwise use weighted average.
     • Round to the nearest integer in [0-100].

3. Produce a strictly formatted JSON output with:

     • 'final_score'
     • 'summary_en' (≤120 characters English concise explanation including key factors and evidence)
     • 'chosen_experts', 'per_model' breakdown (score, weight, specialty match, notes)
     • 'evidence' (keyframe references, detected issues like banding/freeze, MaxVQA factors)
     • 'diagnostics' (score range, fusion method used, routing reasons, suggested next actions)
     • 'confidence' ∈ [0,1]

**Expert model cards (for routing logic)**:
• **COVER**: Uses three parallel branches|technical (Swin Transformer), aesthetic (ConvNet), semantic
  (CLIP encoder)|combined via a cross-gating block to capture compression artifacts, aesthetic
  composition, and semantic coherence (real-time,  96 fps) CVF Open Access.

• **DOVER++**: Extension of DOVER family; overlaps with COVER's aesthetic/technical branches; use
  primarily as a consistency reference unless its output aligns much better with most other experts.

• **UVQ**: Google's YouTube-trained universal VQA model built from millions of UGC examples; robust
  baseline when domain unclear or disagreements are large ResearchGate] Google Research.

• MaxVQA (ExplainableVQA)**: CLIP-based, language-prompted model that provides both overall quality
  and fine-grained human-readable factors (e.g.  banding, blur, aesthetic issues); used only for
  explanation and weight hints, not scoring arXiv.

• **ModularBVQA**: Lightweight, modular baseline model suitable for low-latency deployment and serving
  as a fallback; modest sensitivity to capture-induced distortions.

• **T2VQA**: Text-to-video alignment model designed for AI-generated content, assessing fidelity
  between textual prompt and video; upweighted when video is AI-generated and textual condition is
  provided.

**Video type → baseline weight priors**:
• UGC: UVQ 0.25, COVER 0.25, ModularBVQA 0.15, DOVER 0.10, MaxVQA 0.15

• Short-form/social:  DOVER 0.30, COVER 0.30, UVQ 0.20, Modular 0.10, MaxVQA 0.10

• Gaming:  COVER-Technical 0.35, UVQ 0.25, Modular 0.20, MaxVQA 0.10, DOVER 0.05

• AI-Generated:  T2VQA 0.35, COVER 0.20, UVQ 0.15, MaxVQA 0.15, Modular 0.10, DOVER 0.05

**Weight adjustment formula**:  weight_i = base_i x (1 + 0.5 x specialty_match + 0.3 x agreement_boost +
0.2 x confidence_prior - 0.3 x oob_penalty) Normalize weights to sum to 1.  'specialty_match' ∈ {1,0.5,0}
based on video type match.  'agreement_boost' = closeness to trimmed-mean of scores.  'oob_penalty' =
penalize experts insensitive to detected issues or mismatched type.
```

Figure 5: Prompt for VQA with Q-Router (Tier 1) using GPT-4o.

Clips shorter than 8 frames are discarded, and valid clips are padded for context. Finally, the extractor selects frames by combining (i) top-$K$ high-probability frames, (ii) farthest-point sampling in HSV space for diversity, and (iii) mandatory shot-boundary frames.

HysteresisClips (Algorithm 4) enforces temporal stability when grouping frames into clips. A clip begins when the artifact probability exceeds $\tau_{high}$, and it ends once the probability falls below $\tau_{low}$. This prevents frequent toggling due to small fluctuations. Clips shorter than a minimum length $L_{min}$ are discarded, while valid clips are extended by $P$ frames at both ends to add temporal context.

DiversifiedSelection (Algorithm 5) ensures that sampled frames are both representative of artifacts and diverse in content. First, the top-$K$ frames by probability are selected to guarantee artifact coverage. Next, farthest-point sampling in HSV space ensures visual diversity by spreading samples across color distributions. Finally, all detected shot boundaries are added to preserve scene structure. The union of these sets constitutes the final frame subset, balancing probability, diversity, and structural cues.

The artifact localization pipeline (Algorithm 6) integrates probabilistic sampling, VLM-based filtering, perceptual difference analysis, and structured visualization to identify artifact-prone regions in video content. It proceeds in three main stages:

Figure 6: Prompt for viqual question answering with Q-Router (Tier 0) using GPT-4o.

1. **Probabilistic Frame Extraction.** Each video $\mathcal{V}$ is first processed by the Probabilistic Extractor $\mathrm{PE}(\cdot)$, which assigns frame-wise artifact probabilities based on handcrafted features and logistic weighting. High-probability frames are grouped into clips using the HysteresisClips algorithm with thresholds $\tau_{\mathrm{high}}, \tau_{\mathrm{low}}$, enforcing temporal stability and contextual padding $P$. To ensure coverage, the DiversifiedSelection algorithm further selects frames by combining top-$K$ high-probability samples, farthest-point sampling in HSV space for visual diversity, and mandatory inclusion of shot-boundary frames.

2. **VLM-based Artifact Filtering.** Candidate frames are filtered by a vision–language model $\mathrm{VLM}(\cdot)$ under a structured prompt that classifies frames into three artifact categories: (1) visual hallucinations, (2) image artifacts, or (3) AI-generated inconsistencies. Only frames flagged with a non-trivial label are retained, and clips with no flagged frames are discarded.

3. **Motion-Compensated Perceptual Difference Mapping.** Within each retained clip $c$, consecutive frame pairs $(t, t+1)$ are aligned using optical flow Flow (Farnebäck or TV-L1), which warps the second frame $F_{t+1}$ to the reference $F_t$. The spatial variant of the Learned Perceptual Image Patch Similarity metric $\mathrm{LPIPS}_{\mathrm{spatial}}$ is then applied to compute pixel-level perceptual activation maps $H$, which are normalized to $[0, 1]$ to form interpretable heatmaps $\hat{H}$. Clip-level severity is quantified by aggregating heatmap intensities via mean pooling, and the frame pair with the highest severity is selected as the representative artifact instance. For interpretability, heatmaps are colorized and blended with the original frame ($\alpha = 0.5$), and both raw heatmaps and overlays are saved to disk.

The pipeline returns, for each clip, the representative frame pair, associated severity score, and file paths to saved visualizations. All results are summarized in structured JSON output, including clip boundaries, artifact categories, severity measures, and visualization paths. This ensures reproducibility and facilitates integration with the broader VQA-Router framework for expert routing and perceptual quality assessment.

```
You are VQA-Orchestrator, an expert-level agent for answering user questions about video quality based
on the results from multiple Video Quality Assessment (VQA) experts.

**Your task**:
1. Given the video's sampled frames (16 for single video, 8+8 for comparison) and metadata (type,
   description, orientation, etc.), first classify the video into one of: user generated content(UGC),
   AI generated content (AIGC), computer graphics (CG) Videos.

2. Based on the given expert scores (all already scaled to [0-100]), dynamically assign weights to each
   expert based on their known biases, the video context, inter-model agreement, and confidence priors.
   If metadata/content indicate AI-generated, strongly prioritize **T2VQA**. Use MaxVQA factors to
   downweight unreliable visual regions (e.g., heavy blur or freeze) when deciding.

3. Use these weighted scores as background knowledge to route the question to the most relevant VQA
   experts and generate the final answer.

4. Produce a strictly formatted JSON output with:

     • `answer`
     • `summary_en` (≤120 characters English concise explanation including key factors and evidence)
     • `chosen_experts`, `per_model` breakdown (score, weight, specialty match, notes)
     • `evidence` (keyframe references, detected issues like banding/freeze, MaxVQA factors)
     • `diagnostics` (score range, fusion method used, routing reasons, suggested next actions)
     • `confidence` ∈ [0,1]

**Expert model cards (for routing logic)**:
 • **COVER**: Uses three parallel branches|technical (Swin Transformer), aesthetic (ConvNet), semantic
   (CLIP encoder)|combined via a cross-gating block to capture compression artifacts, aesthetic
   composition, and semantic coherence (real-time, 96 fps) CVF Open Access.

 • **DOVER++**: Extension of DOVER family; overlaps with COVER's aesthetic/technical branches; use
   primarily as a consistency reference unless its output aligns much better with most other experts.

 • **UVQ**: Google's YouTube-trained universal VQA model built from millions of UGC examples; robust
   baseline when domain unclear or disagreements are large ResearchGate] Google Research.

 • MaxVQA (ExplainableVQA)**: CLIP-based, language-prompted model that provides both overall quality
   and fine-grained human-readable factors (e.g. banding, blur, aesthetic issues); used only for
   explanation and weight hints, not scoring arXiv.

 • **ModularBVQA**: Lightweight, modular baseline model suitable for low-latency deployment and serving
   as a fallback; modest sensitivity to capture-induced distortions.

 • **T2VQA**: Text-to-video alignment model designed for AI-generated content, assessing fidelity
   between textual prompt and video; upweighted when video is AI-generated and textual condition is
   provided.

**Video type → baseline weight priors**:
 • UGC: UVQ 0.25, COVER 0.25, ModularBVQA 0.15, DOVER 0.10, MaxVQA 0.15
 • Short-form/social: DOVER 0.30, COVER 0.30, UVQ 0.20, Modular 0.10, MaxVQA 0.10
 • Gaming: COVER-Technical 0.35, UVQ 0.25, Modular 0.20, MaxVQA 0.10, DOVER 0.05
 • AI-Generated: T2VQA 0.35, COVER 0.20, UVQ 0.15, MaxVQA 0.15, Modular 0.10, DOVER 0.05
```

Figure 7: Prompt for viqual question answering with Q-Router (Tier 1) using GPT-4o.

## A.4 ADDITIONAL EXPERIMENT RESULTS

In this section, we present additional experimental results on the test split of Q-Bench-Video.
Since the benchmark has only recently been released, we report results on Tier 1 as a reference to
demonstrate the performance of our proposed Q-Router. Table 4 compares Q-Router against several
representative baselines.

As shown, Q-Router achieves the best overall performance, surpassing GPT-4o by nearly 3 points.
It consistently outperforms competing methods across most question types (Yes-or-No, What/How)
and quality concerns (Technical, Temporal, AIGC), highlighting the advantage of our routing-based
design. Notably, Q-Router demonstrates a strong improvement on AIGC-related quality evaluation,
where accurate detection and reasoning remain particularly challenging. These results confirm that
our framework generalizes effectively to the challenging Q-Bench-Video setting.

## A.5 ADDITIONALY ABALATION STUDY

We further conduct an ablation study using **Qwen3-VL-4B** as the routing model. Table 5 reports
the performance across different question types. It can be observed that: ❶ The Q-Router achieves

```
You are VQA-Orchestrator, an expert-level agent for answering user questions about video quality based
on the results from multiple Video Quality Assessment (VQA) experts.

**Your task**:

1. Given the video's sampled frames (16 for single video, 8+8 for comparison) and metadata (type,
   description, orientation, etc.), first classify the video into one of: user generated content(UGC),
   AI generated content (AIGC), computer graphics (CG) Videos.

2. A heat map overlay on the key frames highlights suspicious regions and related artifact
   classifications are already provided. Lighter regions indicate more severe artifacts, and each
   suspicious frame is already classified as one of:

     • Visual hallucinations | objects, people, or elements that appear artificial or out of place.
     • Image artifacts | compression issues, distortions, blurring, pixelation, or unnatural textures.
     • AI-generated inconsistencies | unrealistic lighting, impossible shadows, or distorted anatomy.

   has been provided to you for reference.

3. Based on the artifact localization prior information and expert scores (all already scaled to
   [0-100]), dynamically assign weights to each expert based on their known biases, the video context,
   inter-model agreement, and confidence priors. If metadata/content indicate AI-generated, strongly
   prioritize **T2VQA**. Use MaxVQA factors to downweight unreliable visual regions (e.g., heavy blur
   or freeze) when deciding.

4. Use these weighted scores as background knowledge to route the question to the most relevant VQA
   experts and generate the final answer.

5. Produce a strictly formatted JSON output with:

     • `answer`
     • `summary_en` (≤120 characters English concise explanation including key factors and evidence)
     • `chosen_experts`, `per_model` breakdown (score, weight, specialty match, notes)
     • `evidence` (keyframe references, detected issues like banding/freeze, MaxVQA factors)
     • `diagnostics` (score range, fusion method used, routing reasons, suggested next actions)
     • `confidence` ∈ [0,1]

**Expert model cards (for routing logic)**:

• **COVER**: Uses three parallel branches|technical (Swin Transformer), aesthetic (ConvNet), semantic
  (CLIP encoder)|combined via a cross-gating block to capture compression artifacts, aesthetic
  composition, and semantic coherence (real-time, 96 fps) CVF Open Access.

• **DOVER++**: Extension of DOVER family; overlaps with COVER's aesthetic/technical branches; use
  primarily as a consistency reference unless its output aligns much better with most other experts.

• **UVQ**: Google's YouTube-trained universal VQA model built from millions of UGC examples; robust
  baseline when domain unclear or disagreements are large ResearchGate] Google Research.

• MaxVQA (ExplainableVQA)**: CLIP-based, language-prompted model that provides both overall quality
  and fine-grained human-readable factors (e.g. banding, blur, aesthetic issues); used only for
  explanation and weight hints, not scoring arXiv.

• **ModularBVQA**: Lightweight, modular baseline model suitable for low-latency deployment and serving
  as a fallback; modest sensitivity to capture-induced distortions.

• **T2VQA**: Text-to-video alignment model designed for AI-generated content, assessing fidelity
  between textual prompt and video; upweighted when video is AI-generated and textual condition is
  provided.

**Video type → baseline weight priors**:

• UGC: UVQ 0.25, COVER 0.25, ModularBVQA 0.15, DOVER 0.10, MaxVQA 0.15

• Short-form/social: DOVER 0.30, COVER 0.30, UVQ 0.20, Modular 0.10, MaxVQA 0.10

• Gaming: COVER-Technical 0.35, UVQ 0.25, Modular 0.20, MaxVQA 0.10, DOVER 0.05

• AI-Generated: T2VQA 0.35, COVER 0.20, UVQ 0.15, MaxVQA 0.15, Modular 0.10, DOVER 0.05
```

Figure 8: Prompt for visual question answering with Q-Router (Tier 2) using GPT-4o.

**competitive performance even with a 4B-parameter model**, comparable to the baselines reported
in Table 2. ❷ There is a **consistent upward trend** from Tier0 to Tier2, indicating that deeper routing
improves decision quality. These results demonstrate that the effectiveness of our framework arises
from the **routing design itself**, rather than relying solely on the capabilities of proprietary large
models such as GPT-4o.

## A.6  LLM USAGE STATEMENT

LLMs were used to assist in editing and improving the readability of sections of this manuscript, in-
cluding clarifying language and refining formatting. The conceptual design, technical contributions,
experiments, and analysis were developed entirely by the authors. All reported results, figures, and
claims originate from the authors' own work and have been independently validated.

**Algorithm 2** Tier-2 Routing Strategy: Expert Selection → Scoring → Fusion

---

**Require:** Frames $F$, expert model cards $\mathcal{C}$, expert model set $\mathcal{E}$.

    **Stage 1: Domain Classification & Initial Expert Selection**
1: $v\_type \leftarrow \text{CLASSIFYVIDEO}(F)$                      ▷ UGC / AIGC / CG
2: Initialize baseline priors:
       $W \leftarrow \text{VIDEOTYPEPRIORS}(v\_type)$
3: Identify candidate experts:
       $\mathcal{E}_{\text{sel}} \leftarrow \text{SELECTEXPERTSBYCARD}(v\_type, \mathcal{C})$
                                   ▷ e.g., T2VQA for AIGC, COVER for UGC

    **Stage 2: Score Retrieval from Selected Experts**
4: **for** $e \in \mathcal{E}_{\text{sel}}$ **do**
5:     $S_e \leftarrow \text{RUNEXPERTMODEL}(e)$                  ▷ all scores are scaled to [0,100]

    **Stage 3: Weighting**
6: $(W, \text{conf}) \leftarrow \text{INFERWEIGHTSANDCONFIDENCE}(F, \mathcal{C})$    ▷ VLM infers expert weights and overall
    confidence from frames + model cards
7: Compute pairwise agreement: $A_{ij} = 1 - \frac{|S_i - S_j|}{100}$
8: **for** $e \in \mathcal{E}_{\text{sel}}$ **do**
9:     $\text{conf}_e = \frac{1}{|\mathcal{E}_{\text{sel}}| - 1} \sum_{j \neq e} A_{ej}$
10:     $W_e \leftarrow W_e \cdot (0.5 + 0.5\,\text{conf}_e)$

    **Stage 4: Artificats Loaclization**
11: $\mathcal{A} \leftarrow \text{PARSEHEATMAPS}(H)$              ▷ suspicious regions + artifact labels

    **Stage 5: Final Fusion and Answer Generation**
12: Fused quality score: $S_{\text{fused}} = \sum_{e \in \mathcal{E}_{\text{sel}}} W_e \cdot S_e$
13: answer $\leftarrow \text{GENERATEEXPLANATION}(S_{\text{fused}}, \mathcal{A}, \mathcal{E}_{\text{sel}})$

    **Stage 6: Structured JSON Output**
14: Construct JSON:

    {"answer", "summary_en", "chosen_experts", "per_model", "evidence", "diagnostics", "confidence"}

        **return** JSON

---

```
Please examine the image carefully and determine if it contains any
of the following:
 1. Visual hallucinations | objects, people, or elements that appear
    artificial or out of place.
 2. Image artifacts | compression issues, distortions, blurring,
    pixelation, or unnatural textures.
 3. AI-generated inconsistencies | unrealistic lighting, impossible
    shadows, or distorted anatomy.
If the image falls into one of the above categories, return only the
corresponding number (1, 2, or 3).  If none apply, return "no".

Only output a single number or "no".
```

Figure 9: Prompt for using GPT-4o to filter the frames extracted by Algorithm 3.

A.7    ADDITIONAL ILLUSTRATIONS OF THE ARTIFACTS LOCALIZATION

We provide additional visual demonstrations of artifact localization on AIGC videos in Figures 10 and 11. The illustrations demonstrate that our artifact localization pipeline can accurately identify

---

**Algorithm 3** ProbabilisticExtractor (PE): Frame Scoring, Clip Formation, and Diversified Selection

---

**Require:** Video $\mathcal{V} = \{F_t\}_{t=1}^T$; weights $\mathbf{w}$, bias $b$; thresholds $\tau_{\text{high}} = 0.65$, $\tau_{\text{low}} = 0.5$; min clip length $L_{\min} = 8$; padding $P$; budgets $K_{\text{top}}$ and $K_{\text{fps}}$; binarization for shot boundaries $\mathcal{S}$ (precomputed)

**Ensure:** Selected frames $\mathcal{F}$, clip set $\mathcal{C}$, probabilities $\{(t, p_t)\}$

    **A. Feature Extraction (per frame $t$)**

1: **for** $t = 1, \dots, T$ **do**

2:     *(a) Motion residuals:* $\text{diff\_mean}_t \leftarrow \begin{cases} \frac{1}{HW} \|F_t - F_{t-1}\|_1, & t > 1 \\ 0, & t = 1 \end{cases}$

3:     *(b) Sharpness energy:* $\text{lap\_var}_t \leftarrow \text{Var}(\text{Laplacian}(\text{Gray}(F_t)))$

4:     *(c) Gradient kurtosis:* $\mathbf{g}_t \leftarrow \sqrt{(\partial_x F_t)^2 + (\partial_y F_t)^2}$; $\text{grad\_kurtosis}_t \leftarrow \frac{\mathbb{E}[(\mathbf{g}_t - \mu)^4]}{\sigma^4}$

5:     *(d) Edge density:* $E_t \leftarrow \text{Canny}(\text{Gray}(F_t))$; $\text{edge\_density}_t \leftarrow \frac{\#\{E_t = 1\}}{HW}$

6:     *(e) Color distribution shift (HSV hist. Bhattacharyya):* $h_t \leftarrow \text{HistHSV}(F_t)$; $\text{hist\_dist\_prev}_t \leftarrow \begin{cases} -\ln\left(\sum_i \sqrt{h_{t-1}(i)\, h_t(i)}\right), & t > 1 \\ 0, & t = 1 \end{cases}$

7:     *(f–g) Content priors (optional):* $\text{face}_t \leftarrow \text{FaceScore}(F_t)$; $\text{text}_t \leftarrow \text{TextScore}(F_t)$

8:     $\mathbf{x}_t \leftarrow [\text{diff\_mean}_t, \text{lap\_var}_t, \text{grad\_kurtosis}_t, \text{edge\_density}_t, \text{hist\_dist\_prev}_t, \text{face}_t, \text{text}_t]$

    **B. Robust Normalization (median–IQR)**

9: **for** feature dimension $i = 1, \dots, 7$ **do**

10:     $m_i \leftarrow \text{median}(\{x_{t,i}\}_{t=1}^T)$; $q_i \leftarrow \text{IQR}(\{x_{t,i}\}_{t=1}^T)$; $\tilde{x}_{t,i} \leftarrow (x_{t,i} - m_i) / \max(q_i, \epsilon)$

    **C. Probabilistic Scoring (weighted logistic)**

11: **Default weights**: motion residuals 0.8, hist. distance 1.0, grad. kurtosis 0.6; remaining dims by validation or set to 0 if unused

12: **for** $t = 1, \dots, T$ **do**

13:     $z_t \leftarrow \mathbf{w}^\top \tilde{\mathbf{x}}_t + b$;    $p_t \leftarrow \sigma(z_t) = \frac{1}{1 + e^{-z_t}}$

14: $\mathcal{P} \leftarrow \{(t, p_t)\}_{t=1}^T$

    **D. Clip Formation via Hysteresis (uses Alg. 4)**

15: $\mathcal{C} \leftarrow \text{HYSTERESISCLIPS}(\mathcal{P}, \tau_{\text{high}}, \tau_{\text{low}}, L_{\min}, P)$

    **E. Diversified Frame Selection (uses Alg. 5)**

16: $\mathcal{F} \leftarrow \text{DIVERSIFIEDSELECTION}(\mathcal{P}, \mathcal{C}, K_{\text{top}}, K_{\text{fps}}, \mathcal{S})$

17: **return** $\mathcal{F}, \mathcal{C}, \mathcal{P}$

---

---

**Algorithm 4** HysteresisClips: Temporal grouping of high-probability frames

---

**Require:** Probabilities $\mathcal{P} = \{(t, p_t)\}_{t=1}^T$; high threshold $\tau_{\text{high}}$; low threshold $\tau_{\text{low}}$; minimum clip length $L_{\min}$; padding $P$

**Ensure:** Set of clips $\mathcal{C}$

1: $\mathcal{C} \leftarrow \varnothing$;   active $\leftarrow$ false

2: **for** $t = 1, \dots, T$ **do**

3:     **if** $\neg$active **and** $p_t \geq \tau_{\text{high}}$ **then**

4:         start $\leftarrow t$;   active $\leftarrow$ true

5:     **else if** active **and** $p_t < \tau_{\text{low}}$ **then**

6:         end $\leftarrow t - 1$

7:         **if** end $-$ start $+ 1 \geq L_{\min}$ **then**

8:             $\mathcal{C} \leftarrow \mathcal{C} \cup \{[\max(1, \text{start} - P), \min(T, \text{end} + P)]\}$

9:         active $\leftarrow$ false

10: **return** $\mathcal{C}$

---

spatio-temporal regions responsible for quality degradations, providing interpretable evidence that complements the overall quality scores.

---

**Algorithm 5** DiversifiedSelection: Balancing probability, diversity, and shot boundaries

---

**Require:** Probabilities $\mathcal{P} = \{(t, p_t)\}$; clip set $\mathcal{C}$; top-$K$ budget $K_{\text{top}}$; farthest-point budget $K_{\text{fps}}$; shot-boundary frames $\mathcal{S}$
**Ensure:** Final selected frame set $\mathcal{F}$
1: $\mathcal{F}_1 \leftarrow \text{TopK}(\{(t, p_t)\}, K_{\text{top}})$ ▷ Highest-probability frames
2: $\mathcal{F}_2 \leftarrow \text{FARTHESTPOINTSAMPLINGHSV}(\{t \mid (t, p_t) \in \mathcal{P}\}, K_{\text{fps}})$ ▷ Diversity in color space
3: $\mathcal{F}_3 \leftarrow \mathcal{S}$ ▷ Mandatory shot-boundary frames
4: $\mathcal{F} \leftarrow \mathcal{F}_1 \cup \mathcal{F}_2 \cup \mathcal{F}_3$
5: **return** $\mathcal{F}$

---

---

**Algorithm 6** Artifact Localization Pipeline

---

**Require:** Video $\mathcal{V}$; Probabilistic extractor $\text{PE}(\cdot)$ with features and weights; VLM filter $\text{VLM}(\cdot)$; optical flow method Flow (Farnebäck/TV-L1); LPIPS metric $\text{LPIPS}_{\text{spatial}}$; thresholds $\tau_{\text{high}}, \tau_{\text{low}}$; clip min length $L_{\text{min}}$; padding $P$; mean pooling mode $M$; overlay opacity $\alpha$
**Ensure:** For each clip: representative frame-pair, heatmaps/overlays; JSON summary

    **Step 1: Probabilistic Frame Extraction**
1: $\{(t, p_t)\} \leftarrow \text{PE}(\mathcal{V})$ ▷ Frame-wise artifact probabilities via features + logistic weights
2: $\mathcal{C} \leftarrow \text{HYSTERESISCLIPS}(\{(t, p_t)\}, \tau_{\text{high}}, \tau_{\text{low}}, L_{\text{min}}, P)$ ▷ Clips with temporal padding
3: $\mathcal{F} \leftarrow \text{DIVERSIFIEDSELECTION}(\{(t, p_t)\}, \mathcal{C})$ ▷ Top-$K$, FPS diversity (HSV), shot-boundaries
    **Step 2: Vision-Language Artifact Filtering**
4: **for all** frame $t \in \mathcal{F}$ **do**
5:     $y_t \leftarrow \text{VLM}(\text{frame}_t)$ ▷ $y_t \in \{1\text{:hallucination}, 2\text{:image artifact}, 3\text{:AI inconsistency}, \text{no}\}$
6: $\mathcal{F}^\star \leftarrow \{t \in \mathcal{F} \mid y_t \in \{1, 2, 3\}\}$
7: $\mathcal{C}^\star \leftarrow \text{RESTRICTCLIPS}(\mathcal{C}, \mathcal{F}^\star)$ ▷ Drop clips with no flagged frames
    **Step 3: Motion-Compensated Perceptual Difference Mapping**
8: **for all** clip $c \in \mathcal{C}^\star$ **do**
9:     $\mathcal{P}_c \leftarrow \{(t, t+1) \mid t \in c, t+1 \in c\}$ ▷ Consecutive pairs
10:     $\text{best}(c) \leftarrow \langle \text{sev} = -\infty, \text{pair} = \varnothing, \text{paths} = \varnothing \rangle$
11:     **for all** $(t, t+1) \in \mathcal{P}_c$ **do**
12:         $F_1 \leftarrow \text{frame}_t, \quad F_2 \leftarrow \text{frame}_{t+1}$
13:         $W_2 \leftarrow \text{WARP}(F_2 \rightarrow F_1; \text{Flow})$ ▷ Align $F_2$ to $F_1$ via optical flow
14:         $H \leftarrow \text{LPIPS}_{\text{spatial}}(F_1, W_2)$ ▷ Pixel-wise LPIPS activation map
15:         $\hat{H} \leftarrow \text{NORMALIZE}(H)$ ▷ $[0, 1]$ normalization
16:         $\text{sev} \leftarrow \text{POOL}(\hat{H}; M)$ ▷ Mean Pooling
17:         **if** $\text{sev} > \text{best}(c).\text{sev}$ **then**
18:             $\text{Overlay} \leftarrow \text{BLEND}(F_1, \text{COLORMAP}(\hat{H}), \alpha)$
19:             $\text{paths} \leftarrow \text{SAVE}\left(\hat{H}, \text{Overlay}\right)$
20:             $\text{best}(c) \leftarrow \langle \text{sev}, \text{pair} = (t, t+1), \text{paths} \rangle$
21: **return** $\{\text{best}(c), \text{paths}(c)\}_{c \in \mathcal{C}^\star}$

---

| Method | Overall | Question type | | Quality Concern | | | |
|---|---|---|---|---|---|---|---|
| | | Yes-or-No | What/How | Technical | Aesthetic | Temporal | AIGC |
| InternVL-Chat | 51.11 | 66.02 | 52.13 | 48.42 | 52.73 | 50.59 | 53.12 |
| LLaVA-Next-Video | 48.69 | 61.34 | 45.95 | 49.03 | 60.94 | 46.97 | 49.40 |
| Video-LLaVA | 43.49 | 64.67 | 40.79 | 43.25 | 54.04 | 42.38 | 42.76 |
| LLaVA-OneVision | 51.70 | 61.34 | 53.88 | 49.35 | **64.15** | 50.68 | 44.30 |
| GPT-4o[†] | 56.50 | 60.60 | 52.20 | 54.60 | 62.70 | 53.70 | 53.80 |
| Q-Router *(Tier 1)* | **59.45** | **63.99** | **54.71** | **59.47** | 62.18 | **58.68** | **64.42** |

Table 4: Comparison of methods on the closed-ended questions in Q-Bench-Video `test` split across question types and quality concerns. [†]: As the models continue to evolve, we reproduce and report the results of the GPT-4o model for comparison.

| Method | Overall | Yes-or-No | What-How | Open-ended |
|---|---|---|---|---|
| Q-Router-Tier0 | 50.06 | 61.02 | 45.31 | 42.42 |
| Q-Router-Tier1 | 51.27 | 62.35 | 46.15 | 43.70 |
| Q-Router-Tier2 | 51.42 | 62.91 | 46.34 | 43.16 |

Table 5: Ablation study using Qwen3-VL-4B as the router.

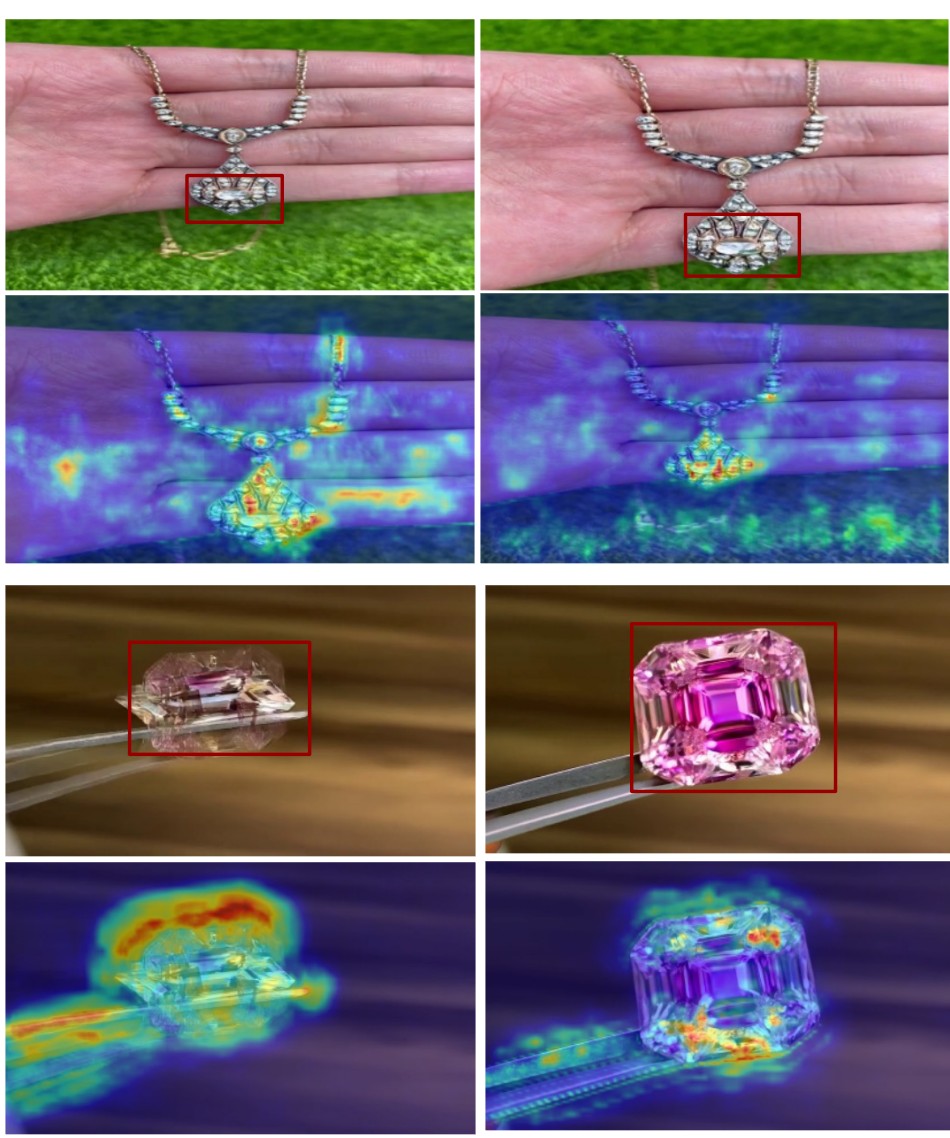

Figure 10: Illustrations of the proposed artifacts localization pipeline on AIGC (Example 1).

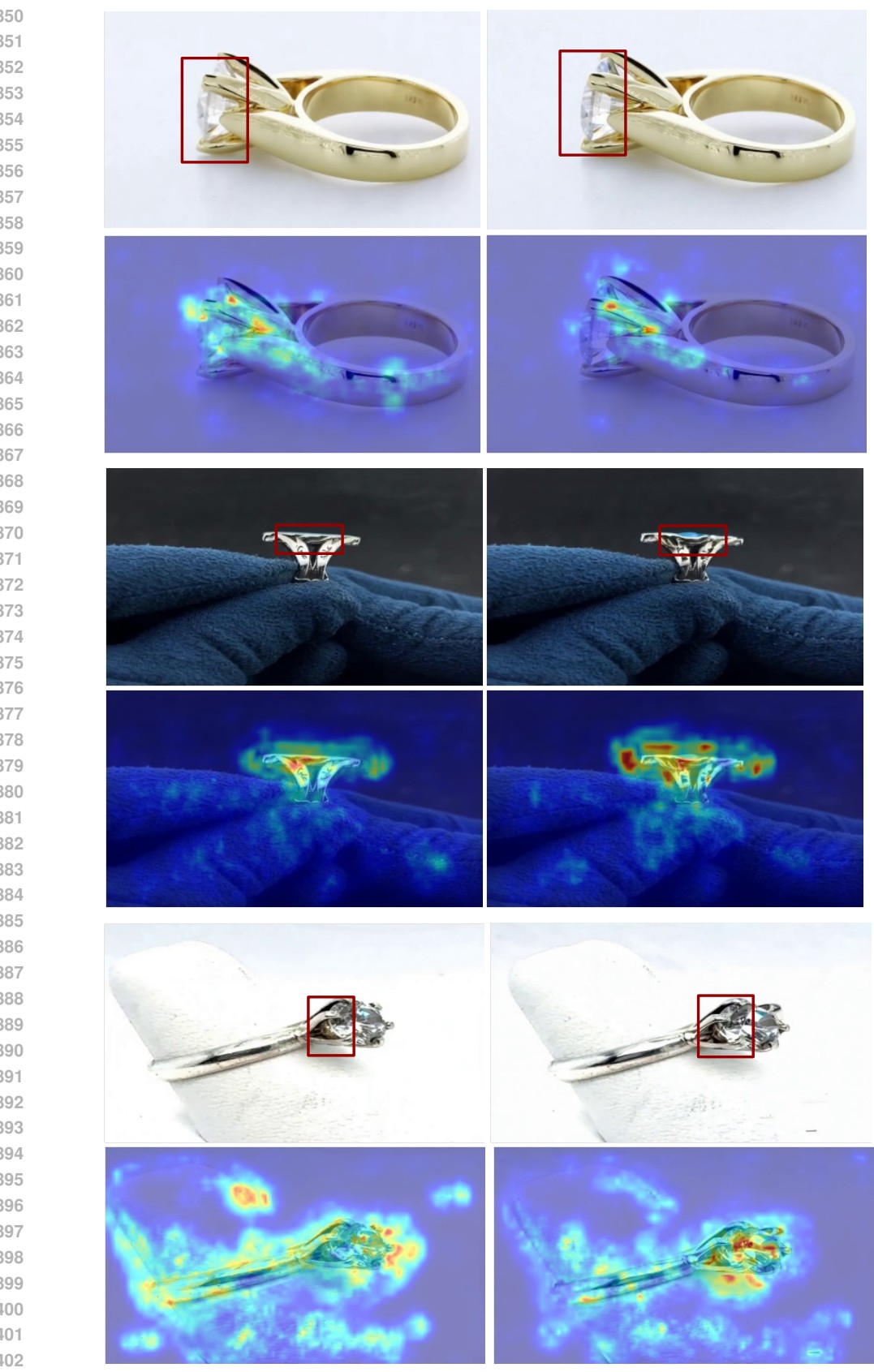

Figure 11: Illustrations of the proposed artifacts localization pipeline on AIGC (Example 2).

