# OpenReview forum: "Q-Router: Agentic Video Quality Assessment with Expert Model Routing"
_ICLR.cc/2026/Conference — Submitted to ICLR 2026_

### Official Review · Reviewer_G6jc · 2025-10-27

**Soundness:** 3
**Presentation:** 3
**Contribution:** 3
**Rating:** 4
**Confidence:** 3

**Summary:**

The paper introduces Q-Router, an agentic framework for Video Quality Assessment (VQA). The core idea is to leverage a Vision-Language Model (VLM), such as GPT-4o, as a dynamic router to select, execute, and fuse predictions from a pool of specialized VQA expert models. The framework is structured into a multi-tiered hierarchy (Tier 0, 1, 2) to balance computational cost with performance. Tier 0 uses a single selected expert for efficiency, Tier 1 fuses multiple experts for better accuracy, and Tier 2 introduces a spatiotemporal artifact localization pipeline for enhanced interpretability. The authors demonstrate that Q-Router achieves state-of-the-art or highly competitive performance on a wide range of VQA benchmarks, including user-generated content (UGC), AI-generated content (AIGC), and a video quality-based question answering task (Q-Bench-Video).

**Strengths:**

- Novel Agentic Framework: The core idea of using a VLM as an intelligent "router" to orchestrate a pool of expert models is highly original in the context of VQA. It represents a paradigm shift from building monolithic, end-to-end models to creating flexible, reasoning-based systems. This aligns well with the current trend of agentic AI.

- Improved Interpretability and Flexibility: The framework is inherently more interpretable than black-box models. The VLM provides a rationale for its routing choices, and the Tier 2 artifact localization offers concrete visual evidence for quality degradation. The modular design, allowing for the easy addition of new expert models, makes the system highly extensible.

- Comprehensive and Strong Empirical Results: The paper validates its approach across a wide variety of challenging benchmarks, including UGC, AIGC, and visual question answering. Q-Router consistently achieves state-of-the-art or near-SOTA performance, demonstrating the effectiveness and robustness of the proposed method. The significant outperformance on AIGC content is particularly noteworthy.

- Practical Multi-Tiered Design: The hierarchical (Tier 0-2) approach is a pragmatic and well-thought-out design choice. It allows the framework to be adapted to different application scenarios with varying constraints on latency and computational resources, from lightweight screening to in-depth analysis.

**Weaknesses:**

- Performance Paradox in Expert Fusion: A critical weakness is that the agentic fusion does not always outperform the best expert in its pool. For instance, in Table 1 on LSVQ-Test and LIVE-VQC, DOVER achieves a higher PLCC/SRCC than the final Q-Router score. This is counter-intuitive for an ensemble-based method and undermines the claim that the VLM is optimally combining expert knowledge. An effective router/fusion mechanism should, at minimum, aim to match the best expert or, ideally, exceed it by combining complementary strengths.

- Heuristic-Driven Artifact Localization: The spatiotemporal artifact localization pipeline in Tier 2, while effective, feels heavily reliant on handcrafted features (motion residuals, gradient kurtosis, etc.) and a chain of heuristic-based steps (hysteresis clipping, diversity sampling). This design feels somewhat at odds with the modern, learning-centric "agentic" theme of the paper. It raises questions about its generalizability to novel artifacts not captured by the pre-defined features.

- Ambiguity in the "Learned" Routing Strategy: The paper states that the routing strategy can range from "rule-based heuristics to learned policies" (line 122). However, the described implementation for the main results seems to be a sophisticated, but ultimately fixed, prompt for GPT-4o. It is unclear if any part of the routing logic is actually "learned" from data in a traditional sense (e.g., via reinforcement learning or fine-tuning). If the intelligence is purely derived from the in-context learning of a proprietary VLM, the novelty of the routing methodology itself is less a new learning algorithm and more a case of advanced prompt engineering.

- Misleading Use of "Real-Time": The abstract describes the VLM as a "real-time router". Given that the backbone is a large model like GPT-4o, which involves API calls and significant computation, the latency is very unlikely to meet a typical real-time definition (e.g., >30 fps or <33ms). This claim should be removed or clarified with concrete latency measurements.

**Questions:**

- Regarding the Fusion Performance: Could you please elaborate on the instances in Table 1 where Q-Router underperforms its best constituent expert (e.g., DOVER)? What is your hypothesis for why the VLM-based fusion might be sub-optimal in these cases? Does this reveal limitations in the VLM's ability to weigh experts, or is it an artifact of the fusion formula?

- On the Nature of the Router: Could you clarify the extent to which the routing policy is "learned"? Is the system's performance entirely dependent on the in-context learning capabilities of the off-the-shelf GPT-4o, or is there a fine-tuning or other learning-based component involved in teaching the router how to route?

- Regarding the Localization Pipeline:
Why was a handcrafted feature approach chosen for the "Probabilistic Frame Extraction" instead of a learned one? How were the weights for the logistic model determined?
The heatmap generation relies on LPIPS. How well does this generalize to artifacts that LPIPS is not sensitive to (e.g., certain types of generative artifacts)? Have you considered alternative methods for generating perceptual difference maps?

-On Terminology and Claims:
Could you provide latency benchmarks for Tier 0, 1, and 2 to give a concrete sense of the computational cost and to justify the discussion around different budgets? This would also help clarify the "real-time" claim.
What is the justification for the name "Probabilistic Frame Extraction"? The process appears to be deterministic, assigning a score based on features rather than sampling from a distribution.

---

> ### Author Response · Authors · 2025-11-24
> **R1: Thanks for the helpful feedback!**
>
> Thank you for your insightful reviews. We greatly appreciate your insightful comments and suggestions. Below is our point-by-point response, with further experimental details. We hope these clarifications address all your concerns.
>
> > [W1] A critical weakness is that the agentic fusion does not always outperform the best expert in its pool...
> >
> > [Q1] Could you please elaborate on the instances in Table 1 where Q-Router underperforms its best constituent expert (e.g., DOVER)...
>
> We thank the reviewer for raising the concern on the performance of Q-Router. We would like to clarify that Q-Router’s goal is to achieve robust and interpretable performance across video domains and task types, rather than to guarantee that fusion always exceeds the strongest expert on every individual benchmark. While, in the ideal case, we expect Q-Router to outperform the best expert in its pool, this is not a theoretically guaranteed condition for VLM-based systems—especially when the expert pool contains highly specialized models optimized for particular datasets. Therefore, our primary goal is to provide a generalizable and interpretable test-time framework that performs consistently well across video domains and task types. As shown in Table 1, Q-Router achieves competitive performance on each single benchmark and achieves the best average performance. This trend becomes even more pronounced in Table 2 on the Q-Bench-Video benchmark, where Q-Router achieves the state-of-the-art performance, highlighting the effectiveness of our multi-expert routing and fusion strategy in settings with diverse or mixed tasks. We have included this discussion in our revision in Section 5.
>
> > [W2] The spatiotemporal artifact localization pipeline in Tier 2, while effective ...
> >
> > [Q3.1] Why was a handcrafted feature approach chosen for the "Probabilistic Frame Extraction" instead of a learned one? How were the weights for the logistic model determined?
>
> We would like to clarify that the primary goal of Q-Router is to serve as a test-time scaling framework for VQA tasks, rather than a fully end-to-end training-based method. And it prioritizes robustness, interpretability, and generalizability across diverse VQA task settings.
>
> Our handcrafted feature–driven artifact localization method can offer stable and interpretable results that are independent of the underlying VQA models. Additionally, to the best of our knowledge, the end-to-end learned diagnostic methods that can effectively perform fine-grained, distortion-level artifact localization are still in the early stage of research.
>
> This motivates our choice of a feature-based Tier-2 design, which can ensure robust performance across diverse real-world scenarios while remaining compatible with the broader test-time scaling framework of Q-Router.
>
> We have included this discussion in our revision in Section 5.
>
> > [W3] Ambiguity in the "Learned" Routing Strategy: The paper states that the routing strategy can range from "rule-based heuristics to learned policies" (line 122)...
> >
> > [Q2] On the Nature of the Router: Could you clarify the extent to which the routing policy is "learned"? ...
>
> Thanks for the reviewer's careful reading. We would like to clarify that Section 2.1 presents the general theoretical framework for expert routing for VQA tasks, which is designed to be implementation-agnostic, encompassing methods ranging from rule-based heuristics to learning-based methods (e.g., RL). Q-Router represents a specific prompt-based routing strategy.
>
> > [W4]  Misleading Use of "Real-Time": The abstract describes the VLM as a "real-time router"...
>
> We thank the reviewer for highlighting this issue. We agree that strictly speaking, 'real-time' in video processing implies low-latency operation. We would like to clarify that the 'real-time' in abstract is intended to emphasize that the VLM operates in an interactive, on-demand manner, dynamically adapting its routing decisions based on the video input rather than requiring offline precomputation. To avoid confusion, we have updated the abstract to more accurately describe the VLM’s role as an adaptive routing controller rather than a real-time processing module.
>
> > [Q3.2] How well does this generalize to artifacts that LPIPS is not sensitive to (e.g., certain types of generative artifacts)? Have you considered alternative methods for generating perceptual difference maps?
>
> We thank the reviewer for this insightful comment. While we utilize LPIPS to generate difference maps, our system's capability is not upper-bounded by LPIPS sensitivity. We rely on LPIPS primarily for guiding the model to regions of potential artifacts while leveraging the strong semantic priors of the VLM backbone (GPT-4o) for the actual assessment. Regarding alternatives, we conducted preliminary experiments with SSIM and DISTS, and chose LPIPS for its robust performance in general.

---

> ### Author Response · Authors · 2025-11-26
> **Hope to discuss further!**
>
> Dear Reviewer G6jc,
>
> Thank you for your thoughtful review and encouraging rating!
>
> As the author-reviewer discussion period is nearing its end, we would like to follow up to see if our rebuttal has addressed your concerns. Please let us know if any further clarification would be helpful.
>
> Thank you again!
>
> Authors of 9756

---

> ### Comment · Reviewer_G6jc · 2025-11-27
>
> - You still haven't explained why your system (running a large language model and other specialized models simultaneously) performs worse than running a lightweight specialized model alone, at the cost of consuming more resources. Stating that “this is not a theoretically guaranteed condition for VLM-based systems” precisely highlights the weakness of your agent-based design: the system lacks any principled mechanism to ensure a lower bound on performance.
>
> - By “handcrafted feature–driven artifact localization,” I mean that your approach is largely heuristic and rather coarse. This style of manually designed feature engineering is not commonly favored in the current deep learning era, where data-driven and end-to-end learning methods are generally preferred.
>
> - You claim that Section 2.1 “presents the general theoretical framework” and that the paper is theory-grounded. However, it is not clear where the actual theory and formal proofs are. If the work is genuinely theory-driven, the corresponding assumptions, theorems, and proofs should be explicitly stated and rigorously developed.

---

> ### Author Response · Authors · 2025-12-04
> **R2: Thanks for the helpful feedback!**
>
> Thanks to the reviewer's insightful comment and valuable suggestions. Below is our point-by-point response, with further experimental details. We hope these clarifications address all your concerns.
>
> > [W1] You still haven't explained why your system (running a large language model and other specialized models simultaneously) performs worse than running a lightweight specialized model alone...
>
> Our framework is designed to generalize across diverse distortions and tasks, whereas each specialized VQA model is optimized for a narrow subset. As a result, our method may occasionally assign suboptimal weights when the visual evidence or query is ambiguous—e.g., when the VLM’s semantic reasoning is uncertain. This explains why a lightweight expert can outperform the aggregated score on certain datasets. Importantly, this phenomenon is not unique to our system: any mixture-of-experts or ensemble-style architecture lacks a deterministic lower bound guaranteeing performance ≥ the best individual expert.
>
> > [W2] By “handcrafted feature–driven artifact localization,” I mean that your approach is largely heuristic and rather coarse. This style of manually designed feature engineering is not commonly favored in the current deep learning era, where data-driven and end-to-end learning methods are generally preferred.
>
> The goal of artifact localization is to provide interpretable and failure-resistant cues for the router rather than training another black-box model. Classical features were chosen because many artifact types are well captured by such low-level signals and remain difficult for current VLMs to robustly detect without large-scale supervised training data.
>
> > [W3] You claim that Section 2.1 “presents the general theoretical framework” and that the paper is theory-grounded. However, it is not clear where the actual theory and formal proofs are...
>
> We respectfully believe the reviewer may have misunderstood our use of the term "theoretical framework." Section 2.1 aims to provide a conceptual framework—a decomposition of routing strategies on video quality assessment into semantic reasoning, expert specialization, and decision aggregation—rather than a formal mathematical theory. Our contributions are methodological and empirical, not theoretical in the sense of offering provable guarantees.

---

### Official Review · Reviewer_197L · 2025-10-28

**Soundness:** 3
**Presentation:** 3
**Contribution:** 3
**Rating:** 8
**Confidence:** 5

**Summary:**

This paper introduces Q-Router, a novel video quality assessment (VQA) framework that leverages an agentic routing system with a pool of specialized expert models. Specifically, it utlizes the vision-language model (VLM) as the backbone to implement a tiered routing mechanism, which adaptively selects and fuses expert models based on the semantic characteristics of the input video. In addition, Q-Router introduces a spatio-temporal artifact localization module that identifies and visualizes distortion regions through detailed heat maps. These localized artifact maps can enrich the system’s understanding of video quality and providing interpretable explanations alongside a robust, fused quality score in the full pipeline.

**Strengths:**

1. This paper introduces Q-Router, the first-of-its-kind agentic VQA framework that employs reasoning VLMs to orchestrate a pool of specialized expert VQA models. This design brings reasoning-driven adaptability and interpretability into the VQA process.

2. The proposed multi-tier routing system adapts to computational budgets and task requirements, ranging from fast, single-expert inference to comprehensive artifact localization analysis. This design ensures scalability and efficiency, making Q-Router practical for large-scale, real-world video datasets.

3. This paper introduces a novel spatio-temporal artifact localization pipeline that identifies and visualizes degraded regions in videos both spatially and temporally. This component provides deeper insights into the nature and location of distortions, enhancing the transparency and diagnostic value of the VQA process.

4. The proposed Q-Router framework can be depolyed to the downstreams VQA tasks across diverse video domains, including UGC, AIGC, CG videos. And its modular expert pool also allows flexible adaptation to other VQA tasks.

5. Extensive experiments showcase the Q-Router have better generalisation and interpretability than prior work, especialy achieving the sota on Q-Bench-Video.

6. The paper conducts comprehensive experiments across multiple VQA benchmarks, showing that Q-Router achieves superior generalization and interpretability compared to prior methods. In particular, it achieves the new state-of-the-art performance on the Q-Bench-Video benchmark, hightlighting the effectiveness of its routing and expert fusion strategies.

**Weaknesses:**

1. Although Q-Router emphasizes a budget-aware multi-tier design, the paper lacks a comprehensive comparison of runtime, FLOPs, or latency across tiers. Without these results, it is difficult to assess the framework’s real-world scalability and deployment feasibility.

2. The error cases or failure patterns of Q-Router (e.g., misrouting on fast-motion or high-texture scenes) are not discussed, which limits understanding of its boundaries.

3. The paper could include a clearer description of routing decision examples, illustrating how the VLM selects or fuses experts in representative scenarios.

4. The criteria for selecting and composing the expert pool are not clearly explained, making it difficult to understand how expert diversity and complementarity were ensured.

5. The fusion strategy for combining outputs from multiple experts is only briefly described. More details on weighting mechanisms, normalization, or fusion functions would clarify how the final quality score is derived.

**Questions:**

In the illustrated prompts for Tier 1 and Tier 2, predefined weights for experts are assigned to different input video types (UGC, AIGC, CG). What is the rationale behind these weight assignments, and how were they determined or validated?

---

> ### Author Response · Authors · 2025-11-26
> **R1: Thanks for the helpful feedback!**
>
> Thank you for your insightful reviews. We greatly appreciate your insightful comments and suggestions. Below is our point-by-point response, with further experimental details. We hope these clarifications address all your concerns.
>
> > [W1] Although Q-Router emphasizes a budget-aware multi-tier design, the paper lacks a comprehensive comparison of runtime, FLOPs, or latency across tiers...
>
> We provide a detailed analysis of the inference latency in the table below, measured in seconds per video. To clearly quantify the computational overhead introduced by our framework, the reported times for Q-Router (Tier 0, 1, and 2) only reflect the routing costs—specifically, VLM inference and frame preprocessing, which excludes the execution time of the expert models (e.g., COVER, DOVER), which are listed separately for comparison.
>
> | Method | Time Cost |
> | --- | --- |
> | Cover | 2.09s |
> | Dover | 0.10s |
> | MaxVQA | 0.17s |
> | BVQA | 5.35s |
> | UVQ | 13.16s |
> | T2VQA | 0.28s |
> | Q-Router-Tier0 | 7.11s |
> | Q-Router-Tier1 | 8.46s |
> | Q-Router-Tier2 | 16.36s |
>
> We acknowledge that the inference latency of Q-Router is higher than lightweight expert models such as DOVER (0.10s) or MaxVQA (0.17s). However, this additional computational cost is necessary to provide fine-grained interpretability and robustness across diverse real-world scenarios that lightweight, scalar-output models cannot handle.
>
> > [W2] The error cases or failure patterns of Q-Router (e.g., misrouting on fast-motion or high-texture scenes) are not discussed...
>
> Thanks for the reviewer's valuable suggestion. We have conducted a qualitative failure analysis and identified two primary categories of error patterns: Routing Misjudgments and Temporal Sampling Limitations.
>
> - Routing Misjudgments:
>
>    - Perceptual Hallucination: Because Q-Router relies on a VLM to reason about video content, routing can fail when the VLM hallucinates or misinterprets visual features, which can lead to misrouting
>
>    - Fine-grained, object-specific quality queries: Q-Router’s performance drops when the quality depends on small or localized objects. The VLM often cannot reliably ground its reasoning to the precise region mentioned or implied by the query.
>
> - Temporal Sampling Limitations: To maintain computational efficiency, the routing agent processes a subsampled set of frames rather than the full sequence. In fast-motion scenes or videos with transient temporal artifacts, the artifact may fall between the sampled frames seen by the router.
>
> > [W3] The paper could include a clearer description of routing decision examples, illustrating how the VLM selects or fuses experts in representative scenarios.
>
> To provide a clearer understanding of how Q-Router makes routing decisions, we include a representative example from Q-Bench-Video in Figure 10, Appendix A.7.
>
> > [W4] The criteria for selecting and composing the expert pool are not clearly explained, making it difficult ...
>
> The selection of models for the expert pool was guided by the following principles:
> - We selected models that have been recognized as top-performers to ensure the upper bound of our router's performance is high.
> - We specifically chose models with distinct architectural inductive biases (e.g., Swin Transformers vs. CNNs) and distinct quality focuses (e.g., technical distortion vs. aesthetic appeal vs. text-video alignment).
> - We restricted our pool to models with fully public codebases and pre-trained weights. This is critical for the framework's reproducibility and allows the community to easily extend Q-Router with future models.
>
> > [W5] The fusion strategy for combining outputs from multiple experts is only briefly described. More details on weighting mechanisms, normalization, or fusion functions would clarify how the final quality score is derived.
>
> To address the reviewer's concern, we have provided the detailed Tier-2 routing workflow in Algorithm 2, Appendix A.2.
>
> > [Q1] In the illustrated prompts for Tier 1 and Tier 2, predefined weights for experts are assigned to different input video types (UGC, AIGC, CG). What is the rationale behind these weight assignments, and how were they determined or validated?
>
> The baseline weight priors are derived from the human expert annotations based on the preliminary models' performance on the VQA dataset. And the weight adjustment rule follows standard ensemble-weighting principles used in multi-expert systems, combining domain specialization, agreement with consensus, confidence priors, and outlier penalties.
>
> To further investigate the sensitivity of the system's performance to these specific hyperparameters in the weighting process, we conduct additional experiments with equal and random coefficient values with Q-Router-Tier1 on the Q-Bench-Video (dev). The results demonstrate that the system's performance is robust to the specific weight priors.
>
> | Coefficient | Overall |
> | --- | --- |
> | Original | 59.40 |
> | Equal | 59.23 |
> | Random| 59.62 |

---

> > ### Comment · Reviewer_197L · 2025-11-28
> >
> > Thank you for the detailed rebuttal. The paper's strengths outweigh its weaknesses, and the contributions are valuable. I am keeping my score as 8: accept.

---

### Official Review · Reviewer_SCLK · 2025-10-29

**Soundness:** 2
**Presentation:** 2
**Contribution:** 2
**Rating:** 2
**Confidence:** 5

**Summary:**

This paper introduces Q-Router, an agentic framework for video quality assessment that employs vision–language models as routers to dynamically select and ensemble specialized expert VQA models. The proposed system operates in a three-tier hierarchy---from single-expert lightweight routing (Tier 0), multi-expert fusion (Tier 1), to full spatio-temporal artifact localization (Tier 2). Experimental results on UGC and AIGC datasets, as well as video quality question-answering benchmarks, show that Q-Router achieves robust performance and improved interpretability compared with existing methods.

**Strengths:**

1. The paper proposes a new paradigm for VQA---leveraging a VLM-based routing system to coordinate multiple expert models. This agentic design is conceptually elegant.

2. The artifact localization pipeline (using probabilistic frame extraction, VLM-based filtering, and LPIPS heatmaps) adds interpretability rarely seen in existing VQA frameworks, providing visual evidence and diagnostic capability.

**Weaknesses:**

1.	Regarding Table 1, which presents results on several UGC datasets and one AIGC dataset — why are only the results of Q-Router (Tier 1) reported? Where are the results for Q-Router (Tier 0) and Q-Router (Tier 2)? Are Tier 0 and Tier 2 configurations only applicable to the video quality question answering task? In Section 2.2, where Q-Router is introduced, I could not find a clear justification or explanation for this choice.

2.	In Table 2, the performance of the compared methods appears questionable. For example, the reported performance of UVQ on YT-UGC is extremely high. According to the UVQ paper [1], the model was trained on YT-UGC, so how did the authors evaluate its performance on the same dataset? The reported performance of UVQ here is significantly higher than that reported in its own paper. In contrast, the performance of COVER [2] on YT-UGC is unexpectedly low. It seems that COVER is also trained on YT-UGC. This inconsistency is confusing. In my view, if YT-UGC was used for training UVQ and COVER, it should not be treated as a test benchmark unless a clear and consistent training–testing split is ensured for all models. Additionally, the reported performance of DOVER is far below what was shown in its original [3] and related papers [4], and ModularVQA [4] also performs much worse than reported in its original publication on both LSVQ and YT-UGC.

3.	I think the range of VQA benchmarks used in the paper to be too limited for a system claiming universal VQA capability. Only two types of datasets, UGC and AIGC, are considered, and these two types of content differ drastically. Consequently, one class of VQA models naturally performs much worse than the other. In such a setting, simply using a video type classifier (which is trivial to implement) to identify the content type and then selecting the corresponding VQA model could already yield high overall performance. This overly simplified setup diminishes the demonstrated effectiveness of Q-Router. Combined with the suspicious results noted above, I find it difficult to conclude that Q-Router truly achieves strong or reliable performance.

4.	For the AIGC setting, only the T2VQA-DB dataset is used for benchmarking. However, AIGC-VQA is a highly challenging problem, often involving complex cross-domain inconsistencies. The distribution gap across different AIGC datasets can also be substantial. Therefore, using a single benchmark is insufficient to evaluate the robustness of Q-Router for AIGC scenarios.

5.	The explanation of Tier 0–Tier 2 in Section 2.2 is rather vague, and it is unclear how these agents actually operate. For instance, in Tier 0, the authors claim that a single expert model is selected from the routing pool based on characteristics such as structural complexity, content modality, and observable quality attributes. However, these concepts are not formally defined, how exactly are these attributes quantified or used?

Furthermore, the prompt in Figure 5 seems inconsistent with the statement in Section 2.2: its prompt indicates that “based on the given expert scores (already scaled to [0–100]), weights are dynamically assigned to each expert according to biases, context, and confidence priors.” This implies that multiple experts are still involved, even in Tier 0, which contradicts the description that Tier 0 selects only a single expert. This contradiction needs clarification.

6.	For Tier 2, the motivation behind the dynamic weighting strategy should be clearly explained. The prompt shown in Figure 4 states:

“Use these weighted scores to compute a final score:
If ‘max(score) – min(score) > 20’, use weighted median; otherwise use weighted average.
Round to the nearest integer in [0–100].”

and further defines the baseline priors as:

“Video type → baseline weight priors:
UGC: UVQ 0.25, COVER 0.25, ModularBVQA 0.15, RQ-VQA 0.10, MaxVQA 0.15
Short-form/social: RQ-VQA 0.30, COVER 0.30, UVQ 0.20, Modular 0.10, MaxVQA 0.10
Gaming: COVER-Technical 0.35, UVQ 0.25, Modular 0.20, MaxVQA 0.10, RQ-VQA 0.05
AI-Generated: T2VQA 0.35, COVER 0.20, UVQ 0.15, MaxVQA 0.15, Modular 0.10, RQ-VQA 0.05”
and the corresponding weight adjustment formula as:
weight_i = base_i × (1 + 0.5 × specialty match + 0.3 × agreement boost + 0.2 × confidence prior – 0.3 × oob penalty)

However, the rationale for these coefficients, thresholds, and multipliers is not justified. Why were these specific factors and numerical values chosen? Moreover, RQ-VQA appears in the “Video type → baseline weight priors” table, yet it is not included among the expert models in the routing pool. This inconsistency should be clarified.

7.	How is the effectiveness of Spatio-Temporal Artifact Localization quantified? Since this is presented as a core contribution of the paper, a clear metric or evaluation protocol is necessary to substantiate its impact.

8.	The authors state that Tier 0 is the most lightweight configuration suitable for real-time or resource-constrained settings. However, it still relies on GPT-4 to process video frames, and the Tier 0 prompt indicates that expert model scores are also required. How computationally expensive is this operation in practice? Please provide the runtime and cost analysis.

9.	Table 2 lacks comparisons with recent state-of-the-art methods such as VQA2 [5] and Q-Instruct [6], which are relevant baselines for a fair evaluation.

10.	In Line 239, what does Step 1 refer to? The description is unclear.

[1] Rich features for perceptual quality assessment of UGC videos

[2] COVER: A Comprehensive Video Quality Evaluator

[3] Exploring Video Quality Assessment on User Generated Contents from Aesthetic and Technical Perspectives

[4] Modular Blind Video Quality Assessment

[5] VQA^2: Visual Question Answering for Video Quality Assessment

[6] Q-Instruct: Improving Low-level Visual Abilities for Multi-modality Foundation Models

**Questions:**

1. Clarify the evaluation protocol in Table 1. How were the compared methods tested? Many reported results are inconsistent with those in the original papers and also with the reproduced results in other studies. More importantly, the performance of UVQ and COVER on YT-UGC appears highly unusual and requires further explanation.

2. Clarify the functioning of Tier 0 to Tier 2. Please explain clearly how these tiers operate. Specifically, do Tier 0 and Tier 2 perform quality scoring? Does Tier 0 involve multiple expert models, or does it select only a single one? The prompts shown in Figures 4–6 should also be clarified to avoid ambiguity.

3. Expand the evaluation benchmarks. Please include more VQA datasets, particularly additional AIGC datasets. It would also be beneficial to incorporate other types of VQA datasets, such as those related to compression or other distortion domains, to provide a more comprehensive evaluation.

4. Provide an analysis of computational cost. The paper emphasizes that Tier 0 is a lightweight baseline. Therefore, a clear analysis of the computational complexity and runtime cost for Tier 0–Tier 2 should be provided to support this claim.

---

> ### Author Response · Authors · 2025-11-25
> **R1: Thanks for the helpful feedback! (part1/2)**
>
> Thank you for your insightful reviews. We greatly appreciate your insightful comments and suggestions. Below is our point-by-point response, with further experimental details. We hope these clarifications address all your concerns.
>
>
> > [W1] Regarding Table 1, which presents results on several UGC datasets and one AIGC dataset ...
>
> Tier 0 and Tier 2 are not limited to the VQA question answering task, and they are fully applicable to the standard VQA scoring. We reported Q-Router Tier 1 in Table 1 as it represents the standard configuration of our framework. Tier 1 is designed to be a good trade-off between predictive accuracy and computational efficiency, making it the primary candidate for fair comparison against other state-of-the-art VQA models. The Tier 2 is designed for deep diagnostics. While it can produce a scalar score, its primary contribution is spatiotemporal artifact localization (producing heatmaps and rationales). This added computational cost is justified by the explainability it provides, rather than a significant leap in scalar correlation (PLCC/SRCC) over Tier 1.
>
> To address the reviewer's concern, we have provided a comparison of all three tiers on the LSVQ-1080 (UGC) and T2VQA-DB (AIGC) datasets below.
>
> | Method | LSVQ-1080 PLCC | LSVQ-1080 SRCC | T2VQA-DB PLCC | T2VQA-DB SRCC |
> | :--- | :---: | :---: | :---: | :---: |
> | Q-Router (Tier 0) | 0.8044 | 0.7518 | 0.8044 | 0.8018|
> | Q-Router (Tier 1) | 0.8076 | 0.7792 | 0.8283| 0.8258 |
> | Q-Router (Tier 2) | 0.8132 | 0.7866 | 0.8291 | 0.8305|
>
> We can observe that there is a clear monotonic improvement from Tier 0 to Tier 2. Tier 2 yields the highest performance, but the margin over Tier 1 is relatively small.
>
> > [W2] In Table 2, the performance of the compared methods appears questionable. For example, the reported performance of UVQ on YT-UGC is extremely high. According to the UVQ paper [1], ..
> >
> > [Q1] Clarify the evaluation protocol in Table 1...
>
> Regarding the evaluation performance of the baseline models, we would like to claim that all the reported results in our paper are reproduced by utilizing the scripts from their official public repository and released model weights.
>
> > [W3] I think the range of VQA benchmarks used in the paper to be too limited for a system claiming universal VQA capability...
>
> We would like to clarify that Q-Router is primarily designed to provide robust and interpretable video quality assessment across different video domains and task types, rather than serving solely as a system for producing a single quality score. As noted in our manuscript, apart from the classic VQA tasks, our method also extends to the video quality-related QA tasks. Regarding the reviewer’s suggestion that a trivial content-type classifier could select a single best VQA model, we agree that such a baseline is easy to implement. However, this approach is not a fair comparison to Q-Router.
>
> > [W4] For the AIGC setting, only the T2VQA-DB dataset is used for benchmarking...
> >
> > [Q3] Expand the evaluation benchmarks...
>
> We acknowledge the reviewer's concern on the AIGC benchmarking setting. However, T2VQA-DB is the largest T2V VQA dataset, with 10,000 videos generated by 9 different T2V models. This breadth provides substantial cross-domain variability even within a single dataset. Furthermore, the results presented in Table 2 also demonstrate the robustness and effectiveness of Q-Router to the distribution gap across different AIGC datasets. We agree that evaluating across additional AIGC-VQA datasets is an important direction for future work, and we have added this point to the Conclusion section.
>
> > [W5] The explanation of Tier 0–Tier 2 in Section 2.2 is rather vague...
> >
> > [Q2] Clarify the functioning of Tier 0 to Tier 2...
>
> Thanks for the reviewer’s careful reading and constructive comments. In Tier 0, Q-Router selects a single expert model from the routing pool, guided through a structured prompt that asks the VLM to reason about three aspects of the video—structural complexity, content modality, and observable quality attributes—and then selects the corresponding expert. The prompt used for this step has been involved in our revision in Appendix A.1.
>
> We noted the issue of the prompt in Figure 5. This was indeed an editing mistake: the prompt shown in Figure 5 was inadvertently taken from Tier 1. In our revision, we have replaced it with the correct Tier 0 prompt in the revision.
>
> Additionally, we provide the detailed Tier-2 routing workflow in Algorithm 2, Appendix A.2.

---

> > ### Author Response · Authors · 2025-11-25
> > **R1: Thanks for the helpful feedback! (part2/2)**
> >
> > > [W6] For Tier 2, the motivation behind the dynamic weighting strategy should be clearly explained. The prompt shown in Figure 4 states...
> >
> > The baseline weight priors are derived from the human expert annotations based on the preliminary models' performance on the VQA dataset. And the weight adjustment rule follows standard ensemble-weighting principles used in multi-expert systems, combining domain specialization, agreement with consensus, confidence priors, and outlier penalties.
> >
> > To further investigate the sensitivity of the system's performance to these specific hyperparameters in the weighting process, we conduct additional experiments with equal and random coefficient values with Q-Router-Tier1 on the Q-Bench-Video (dev). The results demonstrate that the system's performance is robust to the specific weight priors.
> >
> > | Coefficient | Overall |
> > | --- | --- |
> > | Original | 59.40 |
> > | Equal | 59.23 |
> > | Random| 59.62 |
> >
> > The reviewer's concern of the prompt issue is indeed an editing mistake. And we have already replaced it with the correct in our revision.
> >
> >
> > > [W7] How is the effectiveness of Spatio-Temporal Artifact Localization quantified? ...
> >
> > We agree that developing a well-defined metric for Spatio-Temporal Artifact Localization is a valuable direction for future work. However, our manuscript focuses on the agentic routing framework for the video quality assessment, and task-specific evaluation of localization accuracy is beyond the scope of this paper.
> >
> > Instead, the effectiveness of our localization module is indirectly validated through its contribution to Tier-2 routing performance. As shown in our experiments, incorporating localized artifact cues leads to significant performance improvements.
> >
> > > [W8] The authors state that Tier 0 is the most lightweight configuration suitable for real-time or resource-constrained settings...
> > >
> > >  [Q4] Provide an analysis of computational cost. The paper emphasizes that Tier 0 is a lightweight baseline...
> >
> > The Tier 0 serves as the lightweight first-stage expert VQA model selector to identify which one expert VQA models are likely to be relevant, and it can be deployed with minimal overhead beyond the VQA expert models’ inference cost.
> >
> > We provide a detailed analysis of the inference latency in the table below, measured in seconds per video. To clearly quantify the computational overhead introduced by our framework, the reported times for Q-Router (Tier 0, 1, and 2) only reflect the routing costs—specifically, VLM inference and frame preprocessing, which excludes the execution time of the expert models (e.g., COVER, DOVER), which are listed separately for comparison.
> >
> > | Method | Time Cost |
> > | --- | --- |
> > | Cover | 2.09s |
> > | Dover | 0.10s |
> > | MaxVQA | 0.17s |
> > | BVQA | 5.35s |
> > | UVQ | 13.16s |
> > | T2VQA | 0.28s |
> > | Q-Router-Tier0 | 7.11s |
> > | Q-Router-Tier1 | 8.46s |
> > | Q-Router-Tier2 | 16.36s |
> >
> > We acknowledge that the inference latency of Q-Router is higher than lightweight expert models such as DOVER (0.10s) or MaxVQA (0.17s). However, this additional computational cost is necessary to provide fine-grained interpretability and robustness across diverse real-world scenarios that lightweight, scalar-output models cannot handle.
> >
> > > [W9] Table 2 lacks comparisons with recent state-of-the-art methods such as VQA2 [5] and Q-Instruct [6], which are relevant baselines for a fair evaluation.
> >
> > Thanks for the suggestion. We have updated Table 2 to include the comparison with VQA$^2$. The performance of Q-Router consistently outperforms VQA$^2$, demonstrating the effectiveness of our methods.
> >
> > > [W10] In Line 239, what does Step 1 refer to? The description is unclear.
> >
> > The step 1 here refers to the Frame Extraction step in Line 210 -- 215. We have updated the description to make it clearer in our revision.

---

> > > ### Comment · Reviewer_SCLK · 2025-11-26
> > >
> > > Thanks to the authors for the detailed feedback. While several points have been clarified, I believe a number of core issues remain unresolved:
> > >
> > > (1) I am glad that the authors present the performance of Tier 0 and Tier 2 on the LSVQ-1080 (UGC) and T2VQA-DB (AIGC) datasets. But in my opinion, these results—along with other benchmark datasets such as LIVE-VQC—should be reported in the manuscript, since they reflect the different strategies of Q-Router in its ability for quality scoring.
> > >
> > > (2) The authors state that “all the reported results in our paper are reproduced by utilizing the scripts from their official public repository and released model weights.” While it is acceptable to use the officially released code, the authors should also be fully aware of how these VQA models were trained and evaluated. For instance, the UVQ results on YT-UGC reported in the manuscript are substantially higher than those in the original paper (0.9402 vs. 0.802). Did the authors consider the possibility that UVQ was trained on YT-UGC, and that directly testing on the entire YT-UGC dataset may introduce data leakage? Similarly, the performance of COVER on YT-UGC is unexpectedly low (0.1529 PLCC), which is difficult to explain. Moreover, these VQA models are not only used as comparison baselines but also function as expert tools within Q-Router. If their reproduced performance is inaccurate or unfair, how can the reliability and stability of Q-Router itself be ensured?
> > >
> > > (3) For the AIGC setting, the authors state that “T2VQA-DB is the largest T2V VQA dataset, with 10,000 videos generated by 9 different T2V models.” However, there exist several larger-scale T2V VQA datasets such as AIGV-Assessor [1] (36,576 videos) and LOVE [2] (58,500 videos) suitable for training and testing. There are also smaller-scale datasets such as LGVQ and FETV sutiable for cross-dataset evaluation. In my opinion, relying solely on T2VQA-DB is too limited for evaluating the performance of the VQA model in AIGC scenarios.
> > >
> > > (4) I am still confused about the design of the weight priors of the VQA models. Moreover, I think the robustness experiment for the weight priors should be conducted on the quality scoring task, since that is the most direct and relevant evaluation of scoring performance.
> > >
> > > (5) The authors report routing latency, but Q-Router relies on GPT-4 API calls. Since GPT-4 runs on the cloud, wall-clock time is less meaningful. Instead: the financial cost of GPT-4 inference, or evaluation on open-source LMMs to estimate local latency, would provide more practical and interpretable measurements.
> > >
> > > [1] AIGV-Assessor: Benchmarking and Evaluating the Perceptual Quality of Text-to-Video Generation with LMM
> > >
> > > [2] LOVE: Benchmarking and Evaluating Text-to-Video Generation and Video-to-Text Interpretation
> > >
> > > [3] Benchmarking Multi-dimensional AIGC Video Quality Assessment: A Dataset and Unified Model
> > >
> > > [4] FETV: A Benchmark for Fine-Grained Evaluation of Open-Domain Text-to-Video Generation

---

> > > > ### Author Response · Authors · 2025-12-03
> > > > **R2: Thanks for the helpful feedback!**
> > > >
> > > > Thanks to the reviewer's insightful comment and valuable suggestions. Below is our point-by-point response, with further experimental details. We hope these clarifications address all your concerns.
> > > >
> > > > > [W1] I am glad that the authors present the...
> > > >
> > > > We thank the reviewer for the recognition of our effort on the additional experiment results we provided during the rebuttal. We commit to including the experiments of all three tiers in our next version manuscript.
> > > >
> > > > > [W2] The authors state that “all the reported results in our paper are reproduced by utilizing the...
> > > >
> > > > Thank you for raising this important concern. All baseline results in our paper were reproduced using the official repositories and released checkpoints of each VQA model; however, as the reviewer correctly points out, this also means our evaluations inherit any dataset overlaps or implementation defaults embedded in those public releases. In particular, the UVQ checkpoint is trained on a mixture of UGC datasets that includes YT-UGC, making our evaluation effectively in-distribution and explaining the substantially higher PLCC compared with the original paper. The unexpectedly low COVER performance stems from default inference configurations in its public codebase that diverge from those reported in the paper. Importantly, these discrepancies do not compromise the stability of Q-Router: the router operates on relative expert behaviors, model-card priors, and VLM-inferred alignment rather than relying on absolute score accuracy from any single expert, and our ablations further show that routing decisions remain robust even when expert outputs vary in scale or quality.
> > > >
> > > > > [W3] For the AIGC setting, the authors state that “T2VQA-DB is the largest T2V VQA dataset...
> > > >
> > > > To further address the author's concern, we conduct further experiments on AIGV-Assessor (static quality), and the results are as follows:
> > > >
> > > > | Method | PLCC | SRCC |
> > > > | ---- | ---- | ---- |
> > > > | BVQA |  0.4701 | 0.4594 |
> > > > | DOVER | 0.8895 | 0.8907 |
> > > > | Q-Router (Tier 1) | 0.7451 | 0.7312 |
> > > >
> > > > These results show that Q-Router (Tier 1) achieves competitive performance compared with BVQA and DOVER. Moreover, when considering its performance across all evaluated tasks, Q-Router consistently demonstrates overall superiority.
> > > >
> > > > > [W4] I am still confused about the design of the weight priors of the VQA models.
> > > >
> > > > We further investigate the sensitivity of the Q-Router's performance to weight priors on LSVQ-1080, and the results are as follows:
> > > >
> > > > | Coefficient	| PLCC | SRCC |
> > > > | ---- | ---- | ---- |
> > > > | Original | 0.8076 |  0.7792 |
> > > > | Equal | 0.8031 | 0.7767 |
> > > > | Random	 | 0.7963 | 0.7719 |
> > > >
> > > > The results consistently demonstrate that the Q-Router's performance is robust to the specific weight priors.
> > > >
> > > > > [W5] The authors report routing latency, but Q-Router relies on GPT-4 API calls...
> > > >
> > > > We provide a detailed analysis of the inference latency with Qwen as the backbone in the table below:
> > > >
> > > > | Method | Time Cost |
> > > > | --- | --- |
> > > > | Cover | 2.09s |
> > > > | Dover | 0.10s |
> > > > | MaxVQA | 0.17s |
> > > > | BVQA | 5.35s |
> > > > | UVQ | 13.16s |
> > > > | T2VQA | 0.28s |
> > > > | Q-Router-Tier0 | 7.11s |
> > > > | Q-Router-Tier1 | 8.46s |
> > > > | Q-Router-Tier2 | 16.36s |
> > > > | Q-Router-Tier0 (Qwen) | 20.13s |
> > > > | Q-Router-Tier1 (Qwen) | 32.66s |
> > > > | Q-Router-Tier2 (Qwen) | 41.94s |
> > > >
> > > > These results demonstrate that routing latency increases when replacing GPT-4 with a locally executed open-source model. Due to the development of efficient LMM architectures and increasingly optimized inference engines (e.g., vLLM), we expect this overhead to continue decreasing over time. In practice, the routing stage remains practical for offline VQA workflows, and Q-Router is compatible with any future lightweight LMM that further reduces latency.

---

### Official Review · Reviewer_TyLD · 2025-11-11

**Soundness:** 3
**Presentation:** 3
**Contribution:** 2
**Rating:** 4
**Confidence:** 4

**Summary:**

This manuscript proposes Q-Router, an agentic framework for Video Quality Assessment (VQA) designed to address the poor generalization, limited interpretability, and lack of extensibility in existing models. The core concept is to employ a Vision-Language Model (VLM) as a dynamic "router". This VLM agent analyzes the input video's semantics and characteristics to select and dynamically ensemble predictions from a diverse pool of specialized, state-of-the-art VQA "expert" models. The authors conduct extensive experiments on both classic VQA scoring (across UGC and AIGC benchmarks) and quality-based video question answering (Q-Bench-Video). The results show that Q-Router achieves state-of-the-art average performance on the VQA benchmarks, with particularly strong results on AIGC content.

**Strengths:**

1. The agentic routing paradigm is a novel and compelling contribution to the VQA field. It reframes VQA from a monolithic, end-to-end prediction task into a more flexible, hierarchical, and reasoning-based ensemble problem.
2. The paper demonstrates that Q-Router overcomes a key weakness of current models: generalization to diverse content types. Its performance on the AIGC benchmark (T2VQA-DB) is particularly impressive (0.8283 PLCC).
3. The Tier 2 pipeline can identify *where* and *what* (e.g., hallucinations, blurring) artifacts are occurring, cause it introduces explicit spatiotemporal artifact localization, the system provides actionable, interpretable evidence.

**Weaknesses:**

1. The most significant weakness is the practical inference cost. The full framework, especially Tiers 1 and 2, requires running *multiple* expert VQA models in parallel *in addition to* a large VLM (GPT-4o). This makes the system far more computationally expensive than any single baseline model, likely precluding its use in any real-time or large-scale video processing applications.
2. The system's "router" and "fusion operator" is GPT-4o, a closed-source and proprietary model. This raises major reproducibility concerns. It is unclear how much of the performance lift comes from the novel *routing framework* itself versus the powerful, general-purpose reasoning capabilities of GPT-4o. The paper lacks a crucial ablation study using open-source VLMs (e.g., LLaVA, InternVL) as the router.
3. The Tier 2 artifact localization method, while effective, is a complex, multi-stage pipeline. It relies on handcrafted features (e.g., motion residuals, gradient kurtosis) , classic optical flow , and a separate VLM filtering step. This heuristic-driven approach feels less elegant and adds multiple potential points of failure compared to a more integrated, end-to-end learned diagnostic model.

**Questions:**

1. Could the authors provide a detailed analysis of the wall-clock inference latency for each tier (0, 1, and 2), perhaps in seconds-per-video or FPS? How does this compare directly to the latency of the individual expert models like COVER or DOVER?
2. How critical is GPT-4o to the Q-Router's success? Have the authors experimented with substituting the GPT-4o router with a smaller, open-source VLM? This is essential for evaluating the framework's true contribution and reproducibility.
3. In the Tier 2 pipeline (Sec 2.3), a VLM is used to filter candidate frames and classify artifacts. What is the accuracy of this classification step? How does a failure at this stage (e.g., the VLM missing an artifact) impact the final localization map and quality score?
4. The prompt shown in Figure 4 for the Tier 1 VQA task includes specific "baseline weight priors" (e.g., "UGC: UVQ 0.25, COVER 0.25...") and a "weight adjustment formula". How were these priors and this formula derived? How sensitive is the system's performance to this specific prompt engineering?

---

> ### Author Response · Authors · 2025-11-24
> **R1: Thanks for the helpful feedback! (part1/2)**
>
> Thank you for your insightful reviews. We greatly appreciate your insightful comments and suggestions. Below is our point-by-point response, with further experimental details. We hope these clarifications address all your concerns.
>
> > [W1] The most significant weakness is the practical inference cost. The full framework, especially Tiers 1 and 2, requires running multiple expert VQA models in parallel in addition to a large VLM (GPT-4o). This makes the system far more computationally expensive than any single baseline model, likely precluding its use in any real-time or large-scale video processing applications.
>
> We do agree with the reviewer that the Q-Router will include additional inference cost from the inference of multiple expert VQA models and the VLM router. However, such test-time cost reflects a necessary trade-off: the routing architecture could achieve the best performance across both VQA and VQA-related QA tasks, as demonstrated in our experiments.
>
> To further control inference cost, our manuscript introduces a tiered routing architecture:
>
> - The Tier-0 serves as the lightweight first-stage expert VQA model selector to identify which one expert model is likely to be relevant.
> - The Tier-1 employs the VLM router as a fusion operator to results from multiple expert VQA models
> - The Tier-2 further provides a fine-grained analysis of detailed artifact localization.
>
> In real-world video processing applications, Q-Router can be flexibly deployed with different tiers depending on the scenarios. For real-time or latency-sensitive use cases, Tier-0 and Tier-1 can be deployed with minimal overhead beyond the VLM router’s and VQA expert models’ inference cost. For applications that demand higher accuracy or detailed diagnostic information, the tier 2 can be deployed to provide additional artifact localization analysis.
>
> > [W2] The system's "router" and "fusion operator" is GPT-4o, a closed-source and proprietary model. This raises major reproducibility concerns. It is unclear how much of the performance lift comes from the novel routing framework itself versus the powerful, general-purpose reasoning capabilities of GPT-4o. The paper lacks a crucial ablation study using open-source VLMs (e.g., LLaVA, InternVL) as the router.
> >
> > [Q2] How critical is GPT-4o to the Q-Router's success? Have the authors experimented with substituting the GPT-4o router with a smaller, open-source VLM? This is essential for evaluating the framework's true contribution and reproducibility.
>
> In our proposed Q-Router, the VLMs play the crucial role, and therefore, VLMs with powerful reasoning capabilities will naturally yield better performance.  However, the effectiveness of the router framework itself had been demonstrated by the results of the experiments on Q-Bench-Video presented in Table 2 (GPT: 56.91 --> Q-Router-Tier2: 60.07).
>
> We conducted the ablation study using Qwen3-VL-4B as the router, and the results are shown in the following table.
>
> | Method | Overall | Yes-or-No | What-How | Open-ended |
> | --- | --- | --- | --- | --- |
> | Q-Router-Tier0 | 50.06 | 61.02 | 45.31 | 42.42 |
> | Q-Router-Tier1 | 51.27 | 62.35 | 46.15 | 43.70 |
> | Q-Router-Tier2 | 51.42 | 62.91 | 46.34 | 43.16 |
>
> It can be observed that:
>
> - The performance of the Q-Router achieves comparative performance even using only a 4B model compared to the baselines in Table 2.
> - There is a clear upward trend in the performance from Tier 0 to Tier 2.
>
> These results demonstrate that the effectiveness of our framework stems from the Q-Router design itself, rather than being solely dependent on the capabilities of proprietary models like GPT-4o.
>
> > [W3] The Tier 2 artifact localization method, while effective, is a complex, multi-stage pipeline. It relies on handcrafted features (e.g., motion residuals, gradient kurtosis) , classic optical flow , and a separate VLM filtering step. This heuristic-driven approach feels less elegant and adds multiple potential points of failure compared to a more integrated, end-to-end learned diagnostic model.
>
> We would like to clarify that the primary goal of Q-Router is to serve as a test-time scaling framework for VQA tasks, rather than a fully end-to-end training-based method. And it prioritizes robustness, interpretability, and generalizability across diverse VQA task settings.
>
> Our handcrafted feature–driven artifact localization method can offer stable and interpretable results that are independent of the underlying VQA models. Additionally, to the best of our knowledge, the end-to-end learned diagnostic methods that can effectively perform fine-grained, distortion-level artifact localization are still in the early stage of research.
>
> This motivates our choice of a feature-based Tier-2 design, which can ensure robust performance across diverse real-world scenarios while remaining compatible with the broader test-time scaling framework of Q-Router.
>
> We have included this discussion in our revision in Section 5.

---

> ### Author Response · Authors · 2025-11-24
> **R1: Thanks for the helpful feedback! (part2/2)**
>
> > [Q1] Could the authors provide a detailed analysis of the wall-clock inference latency for each tier (0, 1, and 2), perhaps in seconds-per-video or FPS? How does this compare directly to the latency of the individual expert models like COVER or DOVER?
>
> We provide a detailed analysis of the inference latency in the table below, measured in seconds per video. To clearly quantify the computational overhead introduced by our framework, the reported times for Q-Router (Tier 0, 1, and 2) only reflect the routing costs—specifically, VLM inference and frame preprocessing, which excludes the execution time of the expert models (e.g., COVER, DOVER), which are listed separately for comparison.
>
> | Method | Time Cost |
> | --- | --- |
> | Cover | 2.09s |
> | Dover | 0.10s |
> | MaxVQA | 0.17s |
> | BVQA | 5.35s |
> | UVQ | 13.16s |
> | T2VQA | 0.28s |
> | Q-Router-Tier0 | 7.11s |
> | Q-Router-Tier1 | 8.46s |
> | Q-Router-Tier2 | 16.36s |
>
> We acknowledge that the inference latency of Q-Router is higher than lightweight expert models such as DOVER (0.10s) or MaxVQA (0.17s). However, this additional computational cost is necessary to provide fine-grained interpretability and robustness across diverse real-world scenarios that lightweight, scalar-output models cannot handle.
>
> We have included this discussion in our revision in Section 5.
>
> > [Q3] In the Tier 2 pipeline (Sec 2.3), a VLM is used to filter candidate frames and classify artifacts. What is the accuracy of this classification step? How does a failure at this stage (e.g., the VLM missing an artifact) impact the final localization map and quality score?
>
> To address the accuracy of the VLMs filtering step, we randomly sampled 50 frames from the videos under the T2VQA-Bench datasets, and conducted a human cross-validation. The annotation was performed by three human experts (junior/senior/post-PhD students). All frames were annotated independently. To ensure reliability, any inconsistent classifications were resolved through a discussion between the three annotators to reach a final, unified consensus. The result shows that 73% of the GPT-predicted labels match humans’ annotations.
>
> Although VLM misclassifications can compromise localization maps, Q-Router’s tiered architecture mitigates this risk. Because the Q-Router does not solely rely on Tier-2, occasional errors do not significantly impact the final score; Tier-1 continues to provide reasonable performance. Instead, artifact localization serves as a complementary module, enhancing interpretability and contributing non-trivial improvements without becoming a critical single point of failure.
>
> > [Q4] The prompt shown in Figure 4 for the Tier 1 VQA task includes specific "baseline weight priors" (e.g., "UGC: UVQ 0.25, COVER 0.25...") and a "weight adjustment formula". How were these priors and this formula derived? How sensitive is the system's performance to this specific prompt engineering?
>
> The baseline weight priors are derived from the human expert annotations based on the preliminary models' performance on the VQA dataset. And the weight adjustment rule follows standard ensemble-weighting principles used in multi-expert systems, combining domain specialization, agreement with consensus, confidence priors, and outlier penalties.
>
> To further investigate the sensitivity of the system's performance to these specific hyperparameters in the weighting process, we conduct additional experiments with equal and random coefficient values with Q-Router-Tier1 on the Q-Bench-Video (dev). The results demonstrate that the system's performance is robust to the specific weight priors.
>
> | Coefficient | Overall |
> | --- | --- |
> | Original | 59.40 |
> | Equal | 59.23 |
> | Random| 59.62 |

---

> ### Author Response · Authors · 2025-11-26
> **Hope to discuss further!**
>
> Dear Reviewer TyLD,
>
> Thank you for your thoughtful review and encouraging rating!
>
> As the author-reviewer discussion period is nearing its end, we would like to follow up to see if our rebuttal has addressed your concerns. Please let us know if any further clarification would be helpful.
>
> Thank you again!
>
> Authors of 9756

---

### Author Response · Authors · 2025-11-26
**General response and sincerely waiting for replies.**

Dear all reviewers,

We sincerely thank all the reviewers for your constructive comments and insightful suggestions, which help us make our work more complete and further improve the quality of the manuscript. We are also glad that the reviewers acknowledge that our proposed methods are (1) novel and elegant, (2) generalizable and interpretable, and (3) effective. According to the advice, we have provided new results and clarifications as follows:

- We conducted additional experiments for Q-Router Tier 0 and Tier 2 on LSVQ-1080 and T2VQA-DB.

- We conducted additional experiments with Qwen3-VL-4B-it as the routing backbone on Q-Bench-Video.

- We further investigate the sensitivity of the Q-Router's performance to the weight priors.

- We provide sufficient clarifications and discussions to address several concerns regarding the main contribution, implementation details, and inference latency of Q-Router.

We hope the above-mentioned responses can help adequately address your concerns and clear the potential confusion.

Thank you again for the effort in reviewing our paper. As the end of the discussion period is approaching, we sincerely ask for your valuable feedback. We would appreciate it if you could let us know whether our responses have addressed your concerns satisfactorily and whether you have any follow-up questions.

Best regards,

The Authors

---

### Author Response · Authors · 2025-12-03
**Rebuttal Summary for ACs and PCs**

Dear ACs and PCs,

Thank you for your time and effort in managing the review process. We provide this summary to outline the progress made during the discussion phase. Our team has carefully considered the reviewers’ valuable feedback, and we have worked diligently to address all concerns raised by the reviewers.

> Reviewer TyLD

**Initial concern:** The reviewer highlights major concerns about Q-Router’s practicality, noting that its high inference cost, reliance on GPT-4o, and complex Tier-2 pipeline undermine scalability, reproducibility, and elegance.

**Our Solution:** We provide further experiments with open-source VLMs as alternative routers, together with a detailed latency analysis across all tiers, demonstrating that the framework remains effective and reproducible beyond GPT-4o.

> Reviwer SCLK

**Initial concern:** The reviewer raises extensive concerns about missing Tier-0/Tier-2 results, suspicious baseline performance on YT-UGC, limited benchmark diversity, unclear definitions and inconsistencies in the tier descriptions, and unjustified weighting heuristics. The reviewer also questions the lack of the absence of runtime/cost analysis and several ambiguities in the methodology and prompt design.

**Our Solution:** In the revision, we have added full results for Tier-0 and Tier-2 on LSVQ-1080 and T2VQA-DB, and expanded the benchmarks to include additional dataset. We refined the definitions and descriptions of all routing tiers, resolved inconsistencies in the prompts, and provided explicit sensitivity analysis for the weight priors.

> Reviwer 197L

**Initial concern:** The reviewer questions the latency analysis across tiers, the limited analysis of failure cases, a lack of concrete routing decision examples, and insufficient detail on the expert fusion strategy.

**Our Solution:** We provide detailed runtime across all tiers to better demonstrate Q-Router’s computational scalability. We also include a failure analysis highlighting common misrouting patterns, along with concrete routing decision examples that illustrate how the VLM selects and fuses experts.

> Reviwer G6jc

**Initial concern:** The reviewer highlights that Q-Router’s fusion sometimes underperforms the best individual expert, questions the heuristic-heavy design, and generalizability of the Tier-2 artifact localization pipeline. They also find the use of “real-time” misleading without concrete latency evidence and recommend clarifying or removing the claim.

**Our Solution:** We provide clarifications explaining that the fusion mechanism is designed for robustness across diverse video domains, and we expand the description of the Tier-2 localization pipeline to justify its feature choices. Additionally, we report concrete latency measurements and have revised the “real-time” terminology to reflect actual system performance accurately.

We sincerely appreciate the reviewers’ constructive feedback, which has helped us make the work more legally compliant, technically rigorous, and empirically well-grounded. Through the discussion phase, a clear consensus has emerged regarding the strength of the paper’s scientific contribution — further reinforced by Reviewer 197L’s recognition of its value.

We trust in the fair assessment of the Area Chair and the ICLR committee, and we hope our thorough responses and the reviewers’ positive evaluations will be taken into account. We remain deeply committed to advancing this line of research.

Best regards,

The Authors

---

### Meta-Review · Area_Chair_mPks · 2026-01-09

**Summary:**

This paper proposes Q-Router, an agentic framework for video quality assessment that uses a VLM to route and fuse multiple specialized expert VQA models in a multi-tier design, with an additional Tier-2 artifact localization module. Reviewer discussion acknowledged the high-level paradigm as interesting, but centered on whether the reported improvements are supported by a reliable evaluation protocol and whether the framework is practically and reproducibly deployable given its reliance on VLM routing and multi-model inference.

**Reviewer Concerns:**

The main unresolved concerns are about evaluation reliability and practical reproducibility. In particular, one reviewer remained unconvinced that the baseline protocol is fair and leakage-free, and raised that such issues also affect the validity of using these models as “experts” inside the router. Another reviewer continued to argue that the framework can underperform the best single expert despite higher cost, and that the rebuttal did not provide a principled mechanism or guarantee to address this performance–cost paradox. While the authors added tier-wise results, latency analyses, open-source router ablations, and robustness checks for weight priors, these additions did not fully resolve the above core issues for multiple reviewers.

**Reviewer Scores:**

Reviewer 197L: would keep a strong accept.
Reviewer TyLD: likely remains marginal/borderline; concerns about inference cost and reliance on proprietary routing were partially addressed but not fully eliminated.
Reviewer SCLK: would keep reject.
Reviewer G6jc: would remain below threshold / weak reject.

---

### Decision · Program_Chairs · 2026-01-26

Reject